# Multiproxy stable isotope analysis provides insights into diet, animal management, and residential mobility in Old Bara, a metropolitan suburb of the Oyo Empire, West Africa

Moses Oluwaseyi Akogun[1,2]*, Paul Szpak[2], Vicky M. Oelze[3], Kendra Leishman[4], Jay Hilsden[4], Camilla Speller[4], Lisa Janz[1], Jonathan O. Aleru[5], Akinwumi Ogundiran[5,6]*

**1** Department of Anthropology, University of Toronto, Toronto, Ontario, Canada, **2** Department of Anthropology, Trent University, Peterborough, Ontario, Canada, **3** Department of Anthropology, University of California Santa Cruz, Santa Cruz, California, United States of America, **4** Department of Anthropology, University of British Columbia, Vancouver, British Columbia, Canada, **5** Department of Archaeology and Anthropology, University of Ibadan, Ibadan, Oyo, Nigeria, **6** Department of History, Northwestern University, Evanston, Illinois, United States of America

\* moses.akogun@mail.utoronto.ca (MOA); ogundiran@northwestern.edu (AO)

## Abstract

Old Bara was a royal suburb in the metropolis of the Oyo Empire, one of the most important political entities in Africa, ca. 1600–1835. Much of what is known about this town is limited to a few written and oral accounts about its functions as a royal cemetery where the priests and priestesses of the deified kings and other dependents also resided. These limited sources suggest that as a secluded site, Old Bara depended on the palace for its provisions. Archaeological, zooarchaeological, and bioarchaeological research at Old Bara has opened the opportunity to answer new questions about Old Bara's past and to evaluate its identity as a royal site. Here, we use carbon ($\delta^{13}$C), nitrogen ($\delta^{15}$N), sulfur ($\delta^{34}$S), and strontium ($^{87}$Sr/$^{86}$Sr) isotope analysis coupled with zooarchaeology and Zooarchaeology by Mass Spectrometry (ZooMS) to investigate animal provisioning, diet, and residential mobility in Old Bara. Our data provide evidence for local management of animal resources, including the first evidence for horse breeding in the Oyo Empire. The study also challenges the historical accounts that Old Bara was home primarily to palace retirees and priests and priestesses who were required to be celibate and relied on the palace for food provisions. The cumulative evidence from the analyses indicates that the royal town was part of the empire's animal resource production and distribution network.

## Introduction

The archaeology of human-animal relationships in West Africa has employed a range of zooarchaeological methods to investigate the diversity of animal species exploited by people in the past, in order to understand the relationships between hunting and

**Data availability statement:** All relevant data are within the manuscript and its Supporting Information files.

**Funding:** The fieldwork was funded by a National Geographic Explorers Grant NGS-63339R-19 (A.O), an AIA-NEH Grants for Archaeological Research (A.O), and a University of North Carolina at Charlotte Faculty Research Grant (A.O). The analysis was supported via funding from the Canada Research Chairs program (P.S), NSERC Discovery Grant RTI (2023-00124) (P.S), NSERC Discovery Grant (RGPIN-2019-04145) (C.S), Northwestern University's Cardiss Collins Professorship (A.O), Connaught International Scholarship for Doctoral Students (M.O.A), Ontario Graduate Scholarship (M.O.A), and Vanier Canada Graduate Scholarships (FRN: 198885) (M.O.A). There was no additional external funding received for this study. The funders had no role in study design, data collection and analysis, decision to publish, or preparation of the manuscript.

**Competing interests:** The authors have declared that no competing interests exist.

animal husbandry in local contexts, the timing of the introduction of different domesticates into the region, and the role of animals in subsistence strategies and in the formation of complex societies over the past 14,000 years [1–13]. These studies have revealed that animal management was not only for subsistence purposes but also for the pursuit of specialized economic activities, including long-distance trade and the making of ancestors (animals used in transitional death rituals and in maintaining connections with the ancestors) [4,7,8]. This paper contributes to this body of zooarchaeological studies by exploring animal and human relationships in the metropolis of the Oyo Empire in West Africa. For the first time in Nigerian archaeology, we explored the potential of bone collagen by extending our methodological approach beyond traditional zooarchaeological analyses. We use multiproxy stable isotope analysis, including carbon ($\delta^{13}C$), nitrogen ($\delta^{15}N$), sulfur ($\delta^{34}S$), and strontium ($^{87}Sr/^{86}Sr$), alongside Zooarchaeology by Mass Spectrometry (ZooMS), to investigate animal provisioning, diet, and mobility. Whereas previous zooarchaeological studies in Nigeria have analyzed faunal assemblages to understand past environments, subsistence pattern, household formation and social reproduction [1,3,12–15], this study explores what we can learn from biomolecular analysis of faunal remains about how animal resource management was organized and used to forge connections between an imperial metropolis and its provinces, colonies, and trading partners, as well as define and implement the Oyo imperial agenda. We focus on a highly politically charged settlement context—Old Bara, a royal town in the suburbs of the Oyo metropolis, to investigate the patterns and strategies of animal diet and distribution networks that brought animals into the metropolis. In particular, we are interested in what the diets of domestic animals, such as cattle, sheep, goat, donkeys and horses, among others, might reveal about the mobility and local and regional networks of animal exchanges in the metropolis of the Oyo Empire.

## The Oyo Empire

The Oyo Empire was one of the largest political formations in Africa, ca. 1600–1835. At its height, ca. 1775, it stretched between present-day southwest Nigeria and central Togo and from the River Niger to the Atlantic Coast (Fig 1). A few primary documentary sources are available about the empire [16–18]. Most of our knowledge about this political formation originated from oral traditions, and these concentrate on political institutions and economy [19,20]. Archaeological research has enriched our understanding of the settlement landscape of the empire's metropolitan area—Oyo-Ile [14,21] and some of the empire's strategies of territorial expansion, especially through colonization [12].

The Oyo Empire extracted and managed resources from diverse environments, from the woodland savanna to the rainforest and the swampy coastland (Fig 1). As a trading empire, significant resources from across the region were funneled into the metropolis through commerce. Proceeds from tributes, taxes, tolls, and customs also swelled the pockets of the political elite and the palace treasury. As an expansionist state, the empire disrupted the regional demographies and landscapes, as it moved people, animals, plants, and products from one ecological zone to another [12,22].

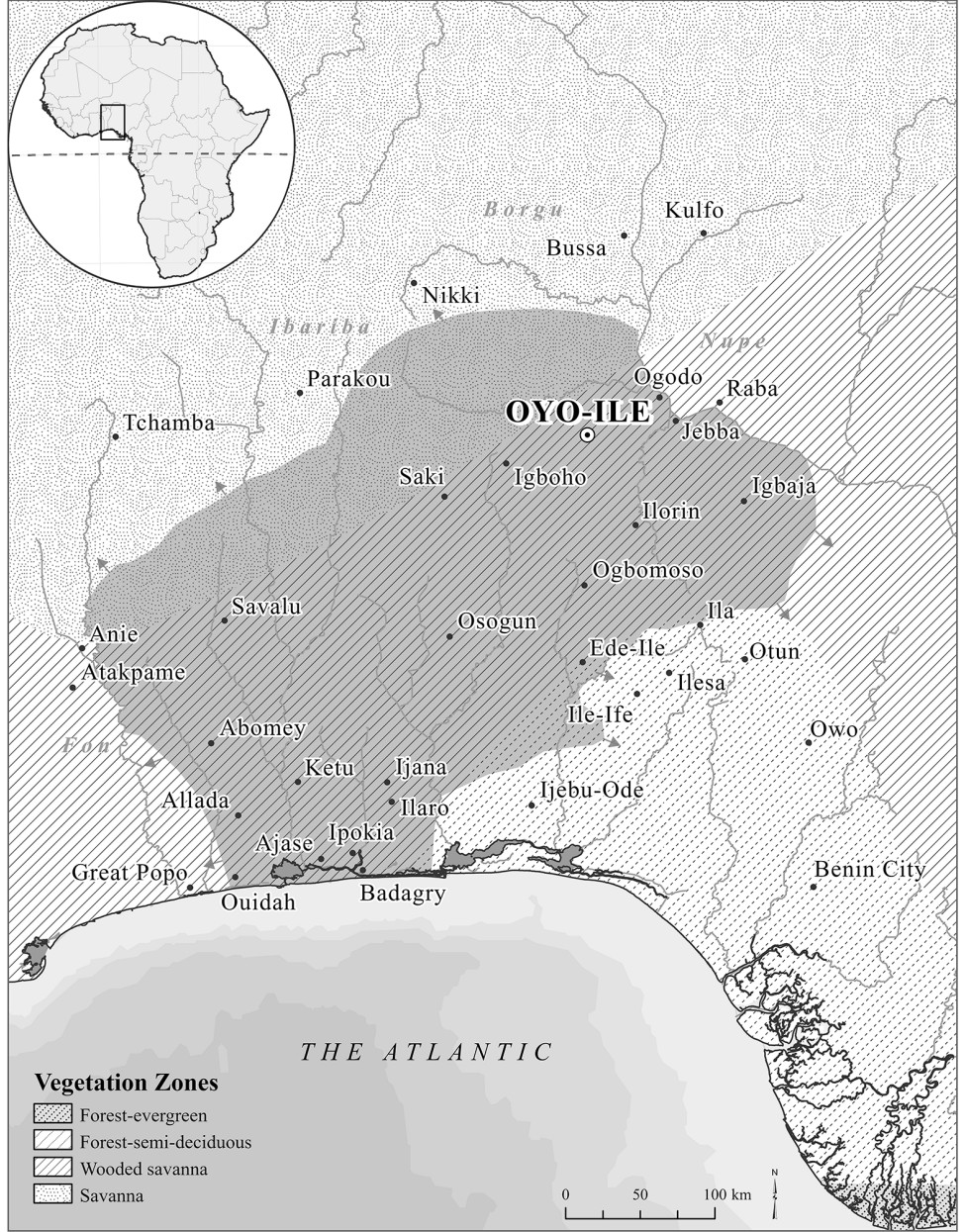

**Fig 1. Map of the extent of the Oyo Empire at its height ca. 1775 showing its capital (Oyo-Ile), some selected towns under the empires and vegetation zones. Map produced by Mech E. Frazier, Geographic Information System (GIS) Specialist, Northwestern University Libraries, specifically for this study (boundary and bathymetry data were sourced from Natural Earth [2025]).**

However, until now, little is known about the diet and animal management in the empire, beyond the limited and generalized information from scanty historical accounts [16,18,19]. There is also a lack of direct archaeological evidence on what people and animals ate, the degree to which the Oyo metropolis harnessed dietary resources locally and from its colonies, vassal states, and trading partners, and how the animal food provisions of the metropolis were managed. This lack of

information creates a significant gap in understanding the political economy of the empire, particularly in how it mobilized labor and governance institutions to secure the food resources necessary to support its expanding bureaucracies, the animal food resources accessed by both state- and non-state-affiliated households and communities, and how this access was mediated by class and socioeconomic status.

The Oyo Empire was a cavalry state, and equines such as horses were used for warfare and performance of prestige, while donkeys were used for transportation [18,23]. In addition, both horses and donkeys were part of the empire's food-ways, especially in elite households, an observation that has been established from both documentary and archaeological sources for the metropolis and one of the empire's colonies, respectively [12,18,22]. Yet, there is no direct archaeological or zooarchaeological evidence for how these animals were obtained and managed. Considering that the empire straddles tropical ecological zones from the woodland savanna in the north to the swamps of the Atlantic littoral in the south, it is often thought that the empire was not an optimal region for equine breeding and maintenance due to the endemic humidity-related diseases, such as trypanosomiasis, that prevail in the woodland savanna and rainforest belts. Therefore, understanding how the empire's political elite obtained their horses and donkeys promises to shed new light on the political economy of the hegemonic state and the economic and social networks it created to access its animal needs, for food, warfare, prestige, politically charged rituals, and haulage/transportation. Our research at Old Bara presents the opportunity to begin to fill this gaping hole on diet and animal management in the metropolis of the Oyo Empire, using multiproxy stable isotope analysis, ZooMS and zooarchaeological analysis.

## Old Bara

Old Bara is located within the wooded savanna ecotone in southwestern Nigeria and lies about 1.2 km northeast of Oyo-Ile, the capital city of the Oyo Empire (Figs 2 and 3). Historical sources identify Old Bara as the royal necropolis where the rituals of coronation and the burial of Oyo kings took place [19]. These sources were not precise as to when Old Bara was first occupied during the Oyo Empire period, but this would not have been later than 1625, fifty years after Oyo-Ile was established as the capital of the emerging empire. Old Bara was evacuated ca. 1835 when the empire collapsed and the capital itself was deserted [24]. During its occupation, Old Bara was heavily fortified by walls, embankments, terraces, and surrounding hills (Fig 3). The perimeter wall of the town measures about 6.6 km in circumference [24]. Administratively, Old Bara was under the leadership of a celibate, the Iyamode. She was a high priestess, one of the most powerful women among the Oyo royal officials, and the only person Oyo kings kneeled to salute [19]. Kings were only allowed to visit this consecrated town only once in their lifetime as part of the coronation rituals [19]. This was an event of pomp and circumstance when the king-elect must visit the tombs of the previous kings from whom he would receive the authority to wear the crown [19]. Upon the king's demise, his remains were returned to Old Bara to be laid to rest. With that return, a temple was built for propitiating the king. Some of the courtiers, including wives, were also relocated to Old Bara to serve as priests and priestesses of the deified king.

According to oral traditions, the town was secluded from outsiders, and most of its residents were also banned from leaving the town [19]. The supposed seclusion of Old Bara and its celibate residents from the rest of the society implies that the aged residents, including retired palace officials and former royal wives, could not produce resources themselves and that the royal town would have relied on the capital for a large part of their provisions. However, the results of the recent excavations challenge the idea that Old Bara was closed off to outsiders [24]. In this study, we use multiproxy stable isotope analysis coupled with data from zooarchaeological analysis to provide answers to the following questions: 1. What was the diet of animals and humans at Old Bara?, and 2. Was the royal town provided with resources from the capital or was it self-sufficient with local resource production? In addition, we used ZooMS to confirm the identification of a few selected faunal remains destined for isotope analysis, given that the assemblage consists of faunal remains that are highly fragmented as a result of intensive culinary processing (see S1 Data for ZooMS results).

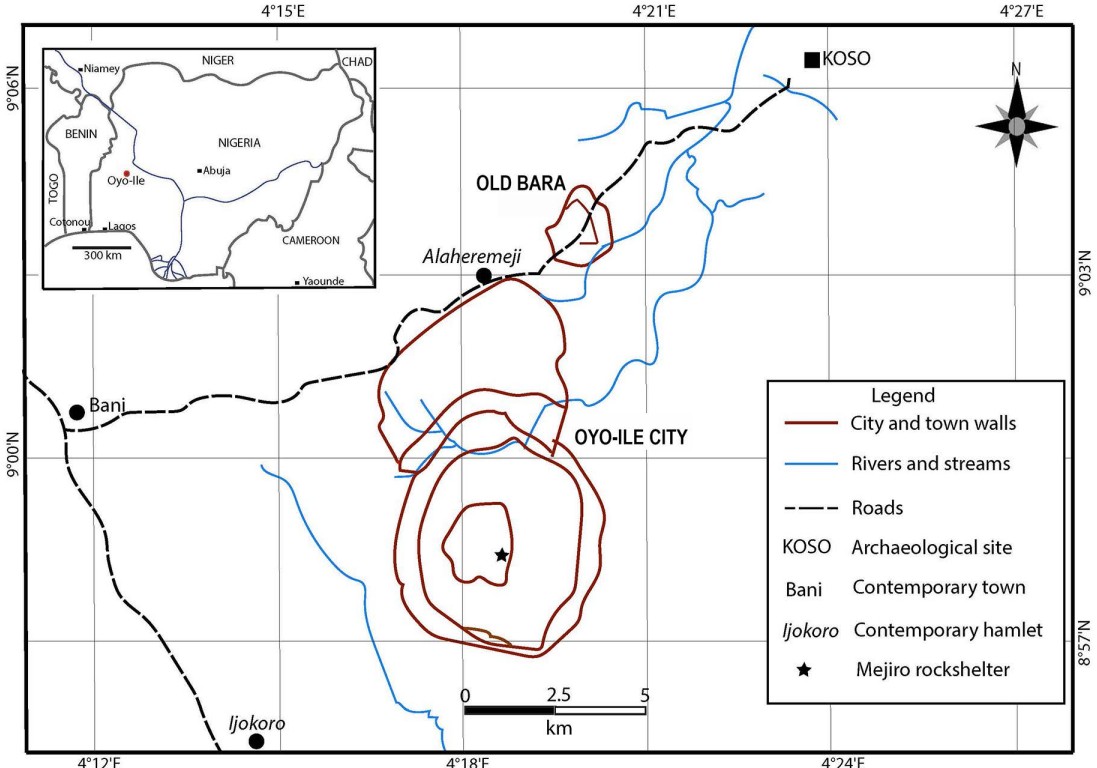

**Fig 2. Map of the metropolitan area of the Oyo Empire showing Oyo-Ile and Old Bara (the royal suburb). Map provided by Akinwumi Ogundiran.**

## Zooarchaeological analysis

Over the last three years, we have carried out detailed zooarchaeological analysis of faunal remains from Oyo-Ile and Old Bara that are in preparation for publication elsewhere. Based on this analysis, we know that the diet of people included both domesticates and a variety of wild species, including horses, cattle, sheep, goat, chicken, oribi, harnessed bushbuck, harte-beest, warthog, rabbit/hare, soft shell turtle, African side-neck turtles, snakes, freshwater fish, freshwater mussels and giant West African snails. Among these species, equids, mostly horses, were among the most important confirming the multiple roles that horses played in the Oyo Empire, according to historical sources [20,23]. These data complement previous faunal data from Oyo-Ile where domesticates dominate the fauna assemblage that also included wild species [14].

Given the importance of horses in the assemblage, it is pertinent to discuss their status, as this is crucial for prove-nience studies. There were two morphotypes of horses in eastern West Africa: the large horses (the Arabian, Dongola, and Barb horses and their large derivatives) and the small horses (West African Dwarf [WAD] horses or the so-called indigenous West African pony). These WAD horses are breeds adapted to the woodland savanna and forest zones of West Africa. The Oyo Empire used both morphotypes [17,25]. Based on morphological identification, the horses analyzed in this study are most consistent with the small morphotypes. Species and size (morphotype) designation were deter-mined using the tooth enamel folding patterns and general tooth conformation. This is made possible because equid tooth formation is controlled by genetics, teeth do not remodel after mineralization, and equids generally show little sexual dimorphism [26–28]. All these reasons make the teeth of equids appropriate for species identification and morphotype designation. However, the postcranial fragments without identification landmarks were not identified to the species level based on size alone, due to the similarity in size between WAD horses and donkeys in this region.

 

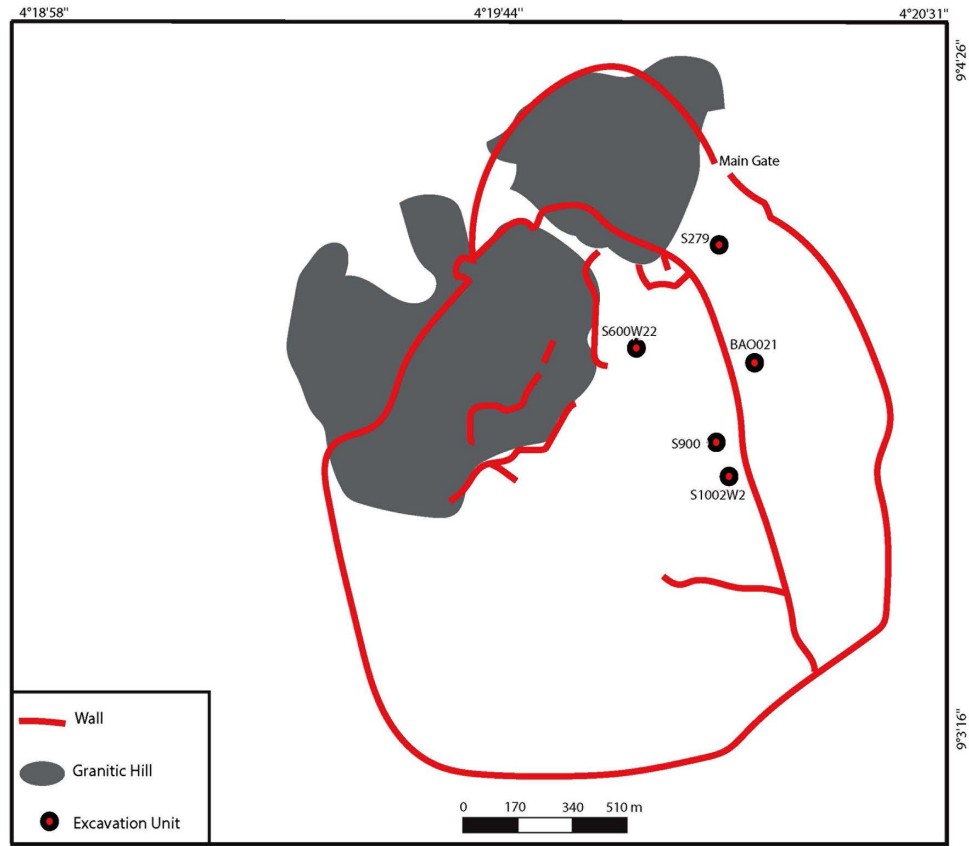

**Fig 3. Map of Old Bara with its fortifications. The five units with materials analyzed in this study are highlighted in red circle. Map provided by Macham Mangut.**

Based on historical sources, these WAD horses, that stand at about 90–110 cm [23,29], often form the Oyo light cavalry or flankers, and the large horses form the heavy cavalry or chargers [25]. Richard Lander, a British explorer, observed that despite the extremely small size of the WAD horses (that he referred to as "Yariba pony"), they were strong and spirited [17]. The tendency to view the large horse morphotypes described in the literature as the ideal cavalry animals is an artifact of the fourteenth century, arising with the development of heavy cavalry [23]. Hence, the use of the term "pony" as a colloquial name for small horses in West Africa has shrouded our understanding of the status, ancestry, and range of these breeds.It is recommended that zooarchaeologists adopt the use of morphotypes when describing horses in West Africa, as some of the large horses could also be regarded as ponies (although they are larger than the WAD horses), given that the tropical wooded environment impedes their optimal growth.

## Isotope background

Stable isotope analysis of animal tissues provides information about the animal's diet and the environment in which it lived by tracing how elements like carbon, nitrogen, sulfur, and strontium move through the food web [30,31]. Animal teeth are like time capsules that record the dietary information of animals during their early life [32,33]. Unlike teeth, animal bones remodel, and they reflect the average diet of a species over periods of years, depending on the type of bone and age at death [34,35]. By analysing the collagen from tooth dentine and bone, and tooth enamel, we can reconstruct the past diet, place of origin, and mobility patterns of animals over specific period in their lives [33,36].

**Carbon.** The carbon isotope composition ($\delta^{13}$C) in animals is a function of the photosynthetic pathways through which the producers (i.e., plants for terrestrial animals) they consume assimilate carbon. Based on these pathways, terrestrial plants are generally grouped into $C_3$, $C_4$, and CAM plants [37]. Much of the savanna in West Africa is dominated by $C_4$ grasses interspersed by $C_3$ trees and shrubs, and the tropical rainforest ecotones are dominated by $C_3$ grasses, shrubs, and trees [38,39]. The proportion of $C_3$ and $C_4$ plants in these environments varies between seasons, and precipitation is the most important factor impacting the distribution of these photosynthetic types [40]. It is possible to estimate the relative proportion of different $C_3$-based and $C_4$-based foods in consumers' diets due to differences in fractionation between different plant species [41]. Economically important $C_4$ plants in West Africa include millets, maize, sorghums, and sugarcane, while wheat, rice, beans, tubers, legumes, and vegetables dominate the often-consumed $C_3$ plants. A wide variety of $C_4$ grasses could also contribute to human diets through grazing animals in this environment. Typically, savanna $C_3$ plants have $\delta^{13}$C values that range between −33 and −22 ‰, and $C_4$ plants have values between −16 and −9 ‰ [31]. The last group of plants (CAM) utilizes a photosynthetic pathway highly adapted to xeric conditions in which $CO_2$ is taken up at night, stored in vacuoles as malate, and fixed during the day. These plants have $\delta^{13}$C values that typically sit between $C_3$ and $C_4$ plants [42]. They include tropical epiphytes and desert succulents [42,43]. Herbivores often have bone collagen $\delta^{13}$C values that are ~4–5 ‰ more positive than those of their whole diet [44].

**Nitrogen.** Plants' nitrogen isotope composition ($\delta^{15}$N) is determined primarily by the mechanism they use to incorporate nitrogen and the $\delta^{15}$N of available mineralized nitrogen [45]. Plants such as legumes that form associations with bacteria that fix $N_2$ from the atmosphere have $\delta^{15}$N values close to 0 ‰, reflecting that of atmospheric nitrogen. Other plants that incorporate nitrogen from nitrogenous compounds such as ammonium ($NH_4^+$), nitrate ($NO_3^-$), and free amino acids often have more positive $\delta^{15}$N values [46], although plants that form symbiotic associations with ectomycorrhizal fungi (common in temperate and boreal forests) or ericoid mycorrhiza (common in bogs and areas with low mineralization rates) tend to have $\delta^{15}$N values lower than leguminous plants [47]. Other environmental and cultural factors, such as aridity, agricultural intensification, manuring, nursing, and the consumption of marine or freshwater resources, can also lead to high $\delta^{15}$N values in plants and animals [48]. In archaeological and ecological studies, the $\delta^{15}$N of animal tissues is also used to infer the trophic position of animals in the food web, as there is a stepwise increase in $\delta^{15}$N between animals and their diet as they move up the food chain. Consumers are often enriched by about 2–6 ‰ in $\delta^{15}$N relative to their diet for most tissues, including bone collagen [49–51].

**Sulfur.** Sulfur isotope ($\delta^{34}$S) analysis is increasingly becoming a tool used in archaeology and ecology as a biogeochemical tracer of diet and mobility patterns [52–54]. The $\delta^{34}$S of animal tissues can be used to estimate the relative contribution of marine, freshwater, and terrestrial foods to animal diets [52]. This is primarily due to the large variation between $\delta^{34}$S values in ocean and terrestrial/freshwater environments. $\delta^{34}$S for marine sulfate is +20.3 ‰ [55], while in freshwater environments, $\delta^{34}$S values should reflect those of the local geology but could be more variable due to the activities of sulfur-reducing organisms like bacteria and archaea [56]. $\delta^{34}$S in terrestrial environments has been estimated to be between −14 and +17 ‰, with the more positive values signifying considerable input from the marine environment [57]. In terrestrial environments, $\delta^{34}$S is used for provenance and to track mobility because the bioavailable sulfur incorporated by plants varies across different geological areas or at variable distances from the ocean [58, 59] and undergoes little to no fractionation in biological systems [60,61]. Based on predicted spatial distribution of $\delta^{34}$S values in West Africa, areas around the coast tend to have more positive values [62].

**Strontium.** Strontium isotope ratios ($^{87}$Sr/$^{86}$Sr) offer another suite of isotope systems often employed as a geographic tracer due to the geospatial variation in bioavailable strontium within a given area [63,64]. Geological bedrock serves as the primary source of strontium in the ecosystem and, once absorbed by plants during nutrient uptake, is often transferred to humans and animals, where it substitutes for calcium in the bioapatite crystal of their teeth and bones [65,66]. Because strontium undergoes little to no fractionation in biological systems and remains unchanged within the food chain, discriminating between local and non-local populations is made possible as long as there is substantial variation in the $^{87}$Sr/$^{86}$Sr ratios

between the different geospatial areas of interest [64]. In addition, $^{87}Sr/^{86}Sr$ ratios could provide insight into the consumption of marine-based resources and help differentiate coastal and non-coastal dwellers [63]. The $^{87}Sr/^{86}Sr$ ratios for modern ocean water are homogeneous globally and have been estimated at 0.7092 [67], while those in old metamorphic and volcanic rocks may cluster around 0.715 and 0.704, respectively [68]. These differences between coastal and terrestrial environments make $^{87}Sr/^{86}Sr$ ratios an important tool for identifying individuals with marine-based diets or residence in coastal environments.

## Materials and methods

### Materials

The materials in this study originated from archaeological excavations conducted in Old Bara between 2018 and 2022 to answer questions about class and gender relations, household organization, and craft activities. All the human and faunal remains in this study originated from five neighborhoods—S279, BAO021, S600W222, S900, and S1002 (Fig 3). The first four are residential contexts, whereas the last, where most of the material in this study came from, is a refuse context that was originally used as a pit-pottery kiln. The remains of two human individuals were found in the refuse context. These remains include an upper second molar, a left talus, a left hallux metatarsal, and a right calcaneus that all belong to one individual, and a cranial fragment that belongs to another individual. Since no other remains of the individuals were found in this context, we posit that these remains were in secondary deposits. It is likely that these fragments were accidentally removed from their original burial contexts through earth movement (e.g., digging) or sweeping activities and deposited along with other materials in the refuse context. Permits to excavate, export, and analyze the materials described in this study were granted by the Nigerian National Park Service and the Nigeria's National Commission for Museums and Monuments between 2018 and 2022 (S2 Data).

For carbon, nitrogen, and sulfur isotope analysis, 46 individuals (human = 2 [4 bones and an upper second molar], fish = 6, cattle = 4, equine = 9, caprine = 11 [8 bones and 3 teeth], turtle = 4, chicken = 4, warthog = 1, hartebeest = 2, harnessed bushbuck = 2, common duiker = 1, snake = 1, cane rat = 1, Rodentia indet. = 1) were analyzed. For strontium, we analyzed 10 equine teeth (horses = 8, donkey = 2), 2 modern snail shells collected during archaeological excavation, and 1 cane rat. With the exception of the human remains, sample selection was done to minimize the risk of sampling an individual multiple times. The assemblage from which these materials were drawn consists of highly fragmented faunal remains from intensive culinary modification. Therefore, sample selection for strontium isotope analysis was based on various loose teeth with different mineralization events from different dwellings. We report 277 distinct carbon, nitrogen, and sulfur isotope measurements and 95 strontium measurements. To the best of our knowledge, this is the most extensive dataset and the first attempt to use sulfur isotope analysis to investigate animal provisioning in West Africa.

### Methods

**Isotope analysis.** Samples for stable isotope analysis were taken from bone and tooth dentine. The surface of bones was abraded to remove all discoloration using an NSK Ultimate dental lab micromotor equipped with a cutting wheel. 47–412 mg of bone and tooth dentine were demineralized in 0.5 M HCl on an oscillating table at room temperature. After demineralization, samples were rinsed to neutrality with Type I water. Humics were removed through a series of 30-min washes in 0.1 M NaOH in an ultrasonic bath until the supernatant became clear. The samples were again rinsed to neutrality. The insoluble residue was denatured in 0.01 M HCl at 65°C for 36 h. Samples were transferred into pre-weighted 4 mL glass vials using a Pasteur pipette, frozen, and lyophilized for 48 h. For the human tooth, a scalpel was used to slice dentinal collagen from the demineralized molar into fifteen micro slices of $1 \pm 0.2$ mm sections with respect to different anatomical parts of the tooth. The crown was sliced into four (S1–S4), the cervix into five (S5–S9), and the root into six sections (S10–S15). Samples from the root were later consolidated into three sections representing $2 \pm 0.4$ mm intervals to make up for the large sample weight (~3 mg) needed to generate reliable $\delta^{34}S$ measurements (>4 μg of S based on

linearity tests of the instrument). S1 to S9 corresponds with Ci to Ri in the modified Moorrees' tooth developmental stages and represents age $2.5 \pm 0.5$ to $8.5 \pm 0.5$ years. Each sample from this sequence represents the individual's average diet over eight months. S10 to S16 corresponds with the stage $R_{1/4}$ to Rc and represents age $9.5 \pm 0.5$ to $14.5 \pm 0.5$ years [69,70]. Since these series have been merged into three series, each represents an average diet over 20 months. For the caprine tooth, one loph from each tooth was cut, demineralized, and sliced using a scalpel into 1 mm sections.

About 3.0 mg of collagen was weighed into tin caps with 6–12 mg of $WO_3$. $\delta^{13}C$, $\delta^{15}N$, and $\delta^{34}S$ and elemental compositions of the collagen were measured using an Elementar Isoprime visION isotope ratio mass spectrometer coupled to a Vario Isotope Cube elemental analyzer at the Trent University Water Quality Centre. For the sequential caprine teeth, about 0.6 mg of collagen were weighed into tin caps, and the $\delta^{13}C$ and $\delta^{15}N$ and elemental compositions of the collagen were measured using a Nu Horizon isotope ratio mass spectrometer coupled to a EuroVector 3300 elemental analyzer at the Trent University Water Quality Centre. $\delta^{13}C$, $\delta^{15}N$, and $\delta^{34}S$ values were calibrated relative to AIR, Vienna Pee Dee Belemnite (VPDB), and Vienna Canyon Diablo Troilite (VCDT) using: IAEA-S-1 (silver sulfide), IAEA-S-2 (silver sulfide), IAEA-S-3 (silver sulfide), IAEA-CH-7 (polyethylene), USGS25 (ammonium sulfate), USGS40 (glutamic acid), USGS41a (glutamic acid), USGS62 (caffeine), and USGS63 (caffeine). Accuracy and precision were measured using in-house standards, including USGS88 (marine collagen), USGS89 (porcine collagen), SRM-1 (caribou bone collagen), SRM-4 (gluten), SRM-7 (albendazole), SRM-8 (amoxicillin), SRM-9 (water buffalo horn), SRM-14 (polar bear bone collagen), and SRM-26 (GH marine collagen). B2162 (Algae/spirulina) was used for elemental composition calibration.

Precision (u(Rw)) was established as $\pm 0.11$ ‰ for $\delta^{13}C$, $\pm 0.20$ ‰ for $\delta^{15}N$, and $\pm 0.56$ ‰ for $\delta^{34}S$ based on analyses from replicates, including calibration standards, check standards, and sample duplicates. Accuracy, or systemic error (u(bias)), was determined to be $\pm 0.23$ ‰ for $\delta^{13}C$, $\pm 0.45$ ‰ for $\delta^{15}N$, and $\pm 0.69$ ‰ for $\delta^{34}S$, derived from the variation between observed and known $\delta$-values of the check standards. The standard uncertainty ($u_c$) for the measurements of $\delta^{13}C$, $\delta^{15}N$ and $\delta^{34}S$ in the samples was calculated to be $\pm 0.25$ ‰ for $\delta^{13}C$, $\pm 0.49$ ‰ for $\delta^{15}N$, and $\pm 0.89$ ‰ for $\delta^{34}S$ [71]. The standard uncertainty ($u_c$) is slightly higher for sulfur due to the high standard deviation on USGS89 for $\delta^{34}S$ during the run from the Vario Isotope Cube elemental analyzer.

Teeth for strontium isotope analysis were first brushed and ultrasonicated in Type I water until the supernatants became clear and were left to dry in a fume hood. For the human tooth, dental calculus was removed prior to ultrasonication. For all horse teeth, peripheral cementum samples were first collected along the distal or buccal side of the tooth to expose the underlying enamel, and these were saved in a labeled 2 mL microcentrifuge vial. Samples were collected from the internal surface of the snail shells. Before collecting sequential enamel and bulk snail shell samples, ~0.1 mm of enamel was abraded off the surface of both the enamel and the shell. Nine peripheral cementum samples from the equid teeth were later analyzed for $^{87}Sr/^{86}Sr$ to characterize the local geological available Sr values for Old Bara and Oyo-Ile to further understand the local variability within the site.

About 2–7 mg of enamel powder was drilled and collected in a 2 mL microcentrifuge tube. For cementum samples, ~4.2–20.6 mg of powdered cementum was collected in a 3 mL microcentrifuge tube. Cementum samples were vortexed in 0.1 M acetic acid and left to react for 20 minutes (acetic acid was added at a ratio of 20 mg to 1 mL). The acetic acid solution was dried down to obtain the exogenous strontium from the cementum samples. Enamel and cementum samples were dissolved in 3 M $HNO_3$ for strontium column chemistry. Strontium was separated from the sample matrix using an Eichrom strontium resin (50–100 µm) [72], and a blank was included for each set of 23 samples to assess contamination. The column was washed 5 times using 3 M $HNO_3$, and strontium was eluted using 0.05 M $HNO_3$. Strontium isotope ratios were measured using a Nu-Plasma-II multi–collector inductively coupled plasma mass spectrometer (MC-ICP-MS) at the Trent University Water Quality Centre. Fifty measurements were taken per sample for analytical precision. Mass-dependent fractionations were normalized using $^{86}Sr/^{88}Sr$ ratio of 0.1194. $^{87}Sr/^{86}Sr$ ratios were corrected for session drift using a value of 0.710236 for NIST-987. The average value for NIST-987 ran across the first run session was $0.710250 \pm 0.000021$ (1σ, n=21) and $0.710250 \pm 0.000011$ (1σ, n=5) for the second session.

Baseline and geographical assignment of equine strontium isotope data. For this study, local $\delta^{34}S$ baselines were established for Old Bara using juvenile long bones of two rodents and the medullary bone of a chicken. An incisor from one of these rodents, a cane rat (*Thryonomys swinderianus*), snail shells (*Limicolaria flammea*), and cementum from equids teeth were used to establish the local $^{87}Sr/^{86}Sr$ ratios for Old Bara and Oyo-Ile. It is worthy of note that $^{87}Sr/^{86}Sr$ ratios of snails may be influenced by rainfall, soil carbonate, and may reflect $^{87}Sr/^{86}Sr$ ratios of shallow rooted plants [73–75]. Cementum is highly susceptible to diagenesis, and the first leachate from this material should reflect the local $^{87}Sr/^{86}Sr$ ratios of the collection localities [76]. We also included $^{87}Sr/^{86}Sr$ ratios from soil samples from Oyo-Ile, Old Bara, and Ede-Ile (the first colony of the Oyo Empire and an important cavalry town) to predict the possible place of origin of the equines (S4 Data). Soil samples were prepared following protocols described in Wang et al. [77]. Although $^{87}Sr/^{86}Sr$ ratios from soil and plant (bioavailable) samples often show overall consistency [78,79], some studies indicate that this relationship is not necessarily strictly linear [80]. The geographical limit of the probability map was selected based on the hypothetical range of the WAD horses. To generate probability maps for the geographic origin of the Old Bara equids, we applied a predicted strontium isoscape of sub-Saharan Africa (mean values) and its estimated standard error as modelled by Wang et al. [77], in combination with the continuous-surface assignment framework from the R package assignR using the pdRaster function [81] in the computational environment R [82].

AMS radiocarbon dating. The collagen samples destined for AMS radiocarbon dating were subjected to an additional ultrafiltration step using a 30 kDa Amicon Ultra Centrifugal Filter. The filters were cleaned using the Keck Carbon Cycle AMS Facility's protocol [83]. Collagen was extracted along with two radiocarbon standards of known dates, Umingmak whale (Fraction Modern Carbon [FMC]=0.4009±0.0008) and Banks Island muskox (FMC=0.6489±0.0011), and Hollis Mine Mammoth as a radiocarbon blank [84] at Trent University. Samples were sent for graphitization and combustion at the Keck Carbon Cycle AMS Facility at the University of California, Irvine. After stable isotope analysis, aliquots of ~1 mg of collagen for each sample were sent to the Ancient DNA and Proteins (ADαPT) Facility at the University of British Columbia for ZooMS analysis.

ZooMS analysis. ZooMS analysis followed the protocol outlined in Buckley et al. [85] and modified for collagen extracts: 100 μL of 50mM ammonium bicarbonate solution ($NH_4HCO_3$), pH 8.0 (AmBic) was then added to the collagen, and samples were vortexed, centrifuged, and left at ambient temperature overnight, before being incubated for one hour at 65°C. Next, 75 μL of supernatant was transferred to a new Eppendorf tube, 0.4 μg of trypsin was added, and the samples were incubated for three hours at 37°C. Samples were centrifuged, acidified with 1 μL of 5% TFA solution, and purified using 100 μL C18 pipette tips with eluates of 50 μL. One microliter of each sample was spotted in triplicate with calibration standards onto a target plate and subjected to Matrix Assisted Laser Desorption/Ionisation Time of Flight Mass Spectrometry (MALDI-ToF-MS) using a Bruker UltrafleXtreme with a smartbeam-II laser at the University of York. Triplicate spectra were averaged, and taxa were identified based on published collagen peptide markers [85–88].

## Results

### Radiocarbon dating

Samples for radiocarbon dating were prepared from two human bones (a talus and a metatarsal) from the same individual. The radiocarbon dating results were calibrated using OxCal. v4.4.4 [89] based on atmospheric data from Reimer et al. [90]. The dates from each sample fall within either the second half of the seventeenth century or the second half of the eighteenth century at a 95.4% two-sigma confidence level (Table 1 and Fig 4). The dates were

**Table 1. Radiocarbon data from two human remains recovered from S1002E5 (Level 5 and 6 respectively) at Old Bara.**

| UCIAMS # | Sample name | fraction Modern | ± | D$^{14}$C (‰) | ± | $^{14}$C age (BP) | ± |
|---|---|---|---|---|---|---|---|
| 296285 | Szpak24082 | 0.9740 | 0.0014 | −26.0 | 1.4 | 210 | 15 |
| 296286 | Szpak24083 | 0.9728 | 0.0014 | −27.2 | 1.4 | 220 | 15 |

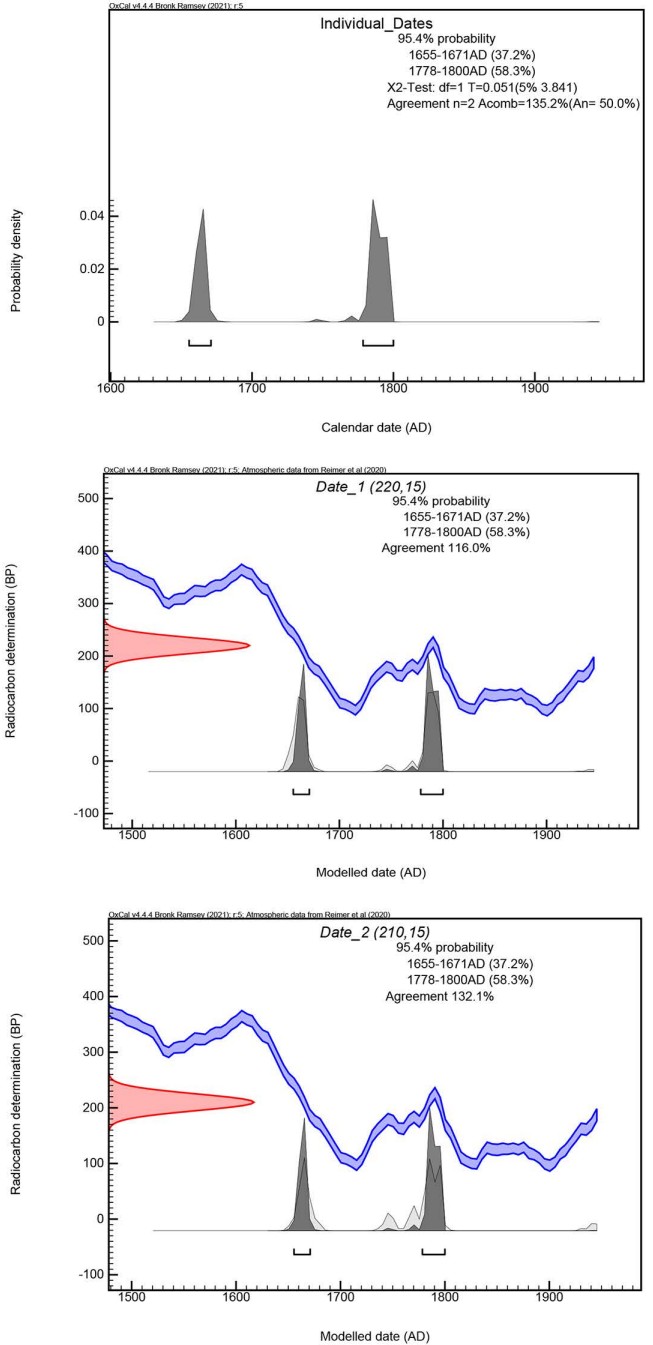

**Fig 4. Modelled dates from Individual 1 from S1002E5, Old Bara (Top: Combined dates using a Bayesian model in OxCal; Middle: Calibrated dates from Sample 296286; Bottom: Calibrated dates from Sample 296285).**

combined using the Bayesian model on OxCal. The results were statistically consistent, with a high agreement index of 135.2%. According to the modeled date, this individual died around 1655–1671AD (37.2%) or 1778–1800AD (58.3%) (Fig 4).

## Isotope analysis

**Collagen quality control and strontium baseline.** The C:N$_{atomic}$ for all samples ranged from 3.13 to 3.35, apart from Samples 27103 and 27344, that had 3.44 and 3.59, respectively (S3 Data). The %C ranged from 14.49% to 44.98%, and the %N ranged from 5.28% to 16.26%. The C:N$_{atomic}$, %C, and %N all fall within the range for the accepted quality control (QC) recommendation for good collagen for archaeological material [91–93]. The %S for the samples ranged from 0.12% to 0.38%, C:S$_{atomic}$ ranged from 189 to 972, and N:S$_{atomic}$ ranged from 58 to 301. While there is currently no generally accepted QC recommendation for sulfur, we access the quality of this data based on observed ranges for modern and archaeological samples [94–96]. Based on these criteria, only 4 samples have %S that is slightly below the lower limit for sulfur, 2 samples have C:S$_{atomic}$ that is higher than the upper limit for sulfur, and 1 sample has N:S$_{atomic}$ that is narrowly higher than the upper limit for sulfur. By comparing the $\delta^{34}$S values from these samples with those of similar taxa in the assemblage, there is no reason to assume sulfur isotope composition has been altered. These QC values may not reflect the preservation of the sulfur isotope composition in the samples [94]. Only Sample 27103 yielded less than 1% collagen (0.80%) but passed other criteria, including proposed sulfur QC [94–96]. Generally, the local $^{87}$Sr/$^{86}$Sr ratios for both Old Bara and Oyo-Ile ranged from 0.7235 to 0.7155. These high ratios are expected for this environment as the sites are on an old Precambrian basement complex (Figs 5 and 6).

**$\delta^{13}$C, $\delta^{15}$N, and $\delta^{34}$S of animal remains.** The $\delta^{13}$C, $\delta^{15}$N, and $\delta^{34}$S values from the faunal remains show a clear separation between wild species and domesticates like equines, cattle, sheep, goats, and chickens (Figs 7 and 8). The $\delta^{13}$C from wild antelopes also differentiated browsers like harnessed bushbuck and common duiker from heavy grazers like hartebeest (Fig 7). Caprines showed higher variability in $\delta^{13}$C and $\delta^{15}$N values compared to other domesticates. Considering the variability in the $\delta^{13}$C and $\delta^{15}$N values from these caprines, causality was investigated by assessing how these isotope systems vary within the teeth of three caprines. $\delta^{13}$C values within a caprine tooth could vary as much as 3.20 ‰, while the degree of variability was less for $\delta^{15}$N, with the highest observed range being 1.70 ‰ (Figs 1 and 2, S1 Data). These variations may stem from seasonal fluctuations in C$_3$ and C$_4$ plants, but similar patterns have also been observed in millets, including those grown under the same conditions in West Africa [97–99]. The $\delta^{34}$S values of the domesticates show that most of the equines, caprines, chickens, and cattle were non-coastal and probably local (Fig 8). A cow/bull, Sample 27223 ($\delta^{34}$S = +16.3 ‰), originated from a coastal environment, as this value reflects foraging in an environment that received sea spray (Fig 8).

We further tested whether one of the local chickens (a hen) was raised locally by comparing the isotope composition of its cortical and medullary bones (Fig 6J). The medullary bone is found only in female egg-laying birds and is a repository for calcium used for eggshell formation [100]. This is a very active bone with a turnover of ~10–15 times faster than cortical bones [101]. Its rapid turnover rate makes it possible to capture the diet of egg-laying chickens within a short time in the egg-laying cycle. For chickens, collagen is deposited for medullary bone formation during the matrix formation phase after each oviposition, which occurs within a day [102]. For the sample analyzed, the difference between the cortical bone (Sample 27287) and medullary bone (Sample 27344) is 0.23 ‰ for $\delta^{13}$C, 0.54 ‰ for $\delta^{15}$N, and 0.55 ‰ for $\delta^{34}$S (S3 Data). These differences are negligible and show that the average lifetime diet and locality of this hen, represented by its cortical bone, is the same as that of its last egg-laying cycle, represented by its medullary bone. Three sequential samples were taken along the tusk for the warthog (S3 Data), and the tight $\delta^{34}$S values (Table 2) within the tooth are not unexpected, as they are non-migratory animals [103].

**$^{87}$Sr/$^{86}$Sr ratios of Equines.** Given that most of the horses in this study are represented by a single tooth, we can not estimate their full mobility pattern covering the full stage of teeth mineralization that is possible when suitable teeth from complete mandibles are available [104]. Generally, enamel mineralization for the deciduous teeth of horses occurs during pregnancy [105]. Therefore, the deciduous teeth analyzed in this study should reflect the diet and mobility history of the broodmare during pregnancy rather than those of the foal. Enamel mineralization in M1 and M2 begins when foals are still

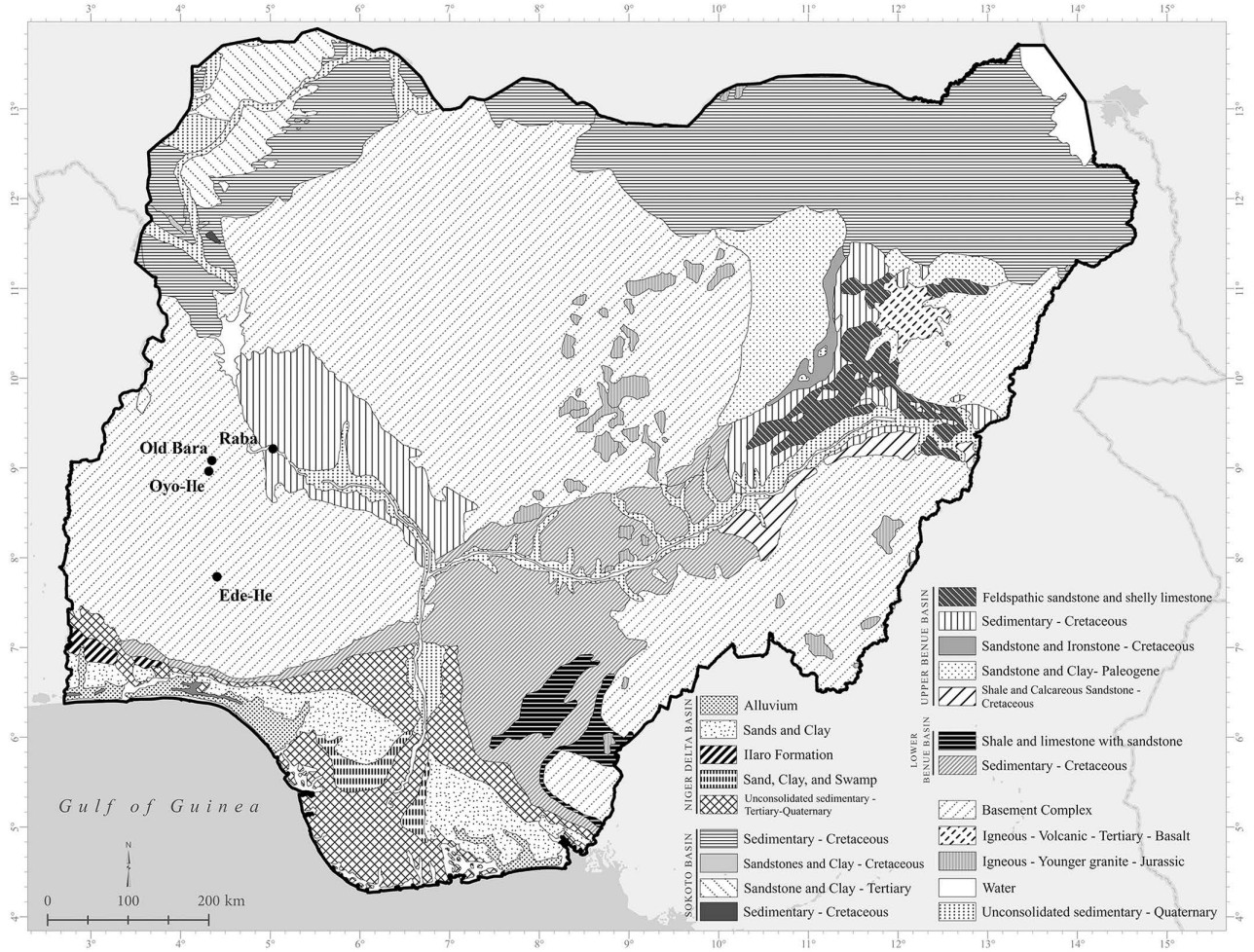

**Fig 5. Geological Map of Nigeria.** Map produced by Mech E. Frazier, Geographic Information System (GIS) Specialist, Northwestern University Libraries, specifically for this study (boundary data were sourced from Natural Earth [2025]; geological boundary data were sourced from the United States Geological Survey [2002]).

nursing, making it possible to identify whether these individuals were locally raised in Old Bara or were brought in. P3 and P4 begin to mineralize 12 months after birth and would reflect mobility patterns from this time [104]. For domestic horses, the P3, P4, M1, M2, and M3 mineralize over an average of 22, 32, 22.5, 30, and 34 months, respectively [104]. Given that the teeth analyzed in this study are only slightly worn, they should capture the mobility of the individuals represented by their P3/P4 and M1/M2 over a period of at least 1.5 years.

Samples from Horses 3 and 4 are deciduous teeth, with the dP3 and dP4 teeth from horse 3 in their early wear stage (Fig 6G). This wear pattern suggests that the estimated age of this individual is around a few weeks from birth, while Horse 4 is around 4–16 months. The presence of these foals in Old Bara, coupled with information from their modeled place of origin, suggests local birth (Fig 9). This is the first evidence of local horse breeding in the Oyo Empire. The alveolar bone attached to these teeth has a $\delta^{34}$S value (Sample 27234) that is slightly below the estimated lower limit for the site, further supporting the notion that this individual is likely local. Sequential $^{87}$Sr/$^{86}$Sr analysis from two other horses also suggests that they were probably local or might have spent substantial time in their early life in Old Bara (Horses 1 and 2)

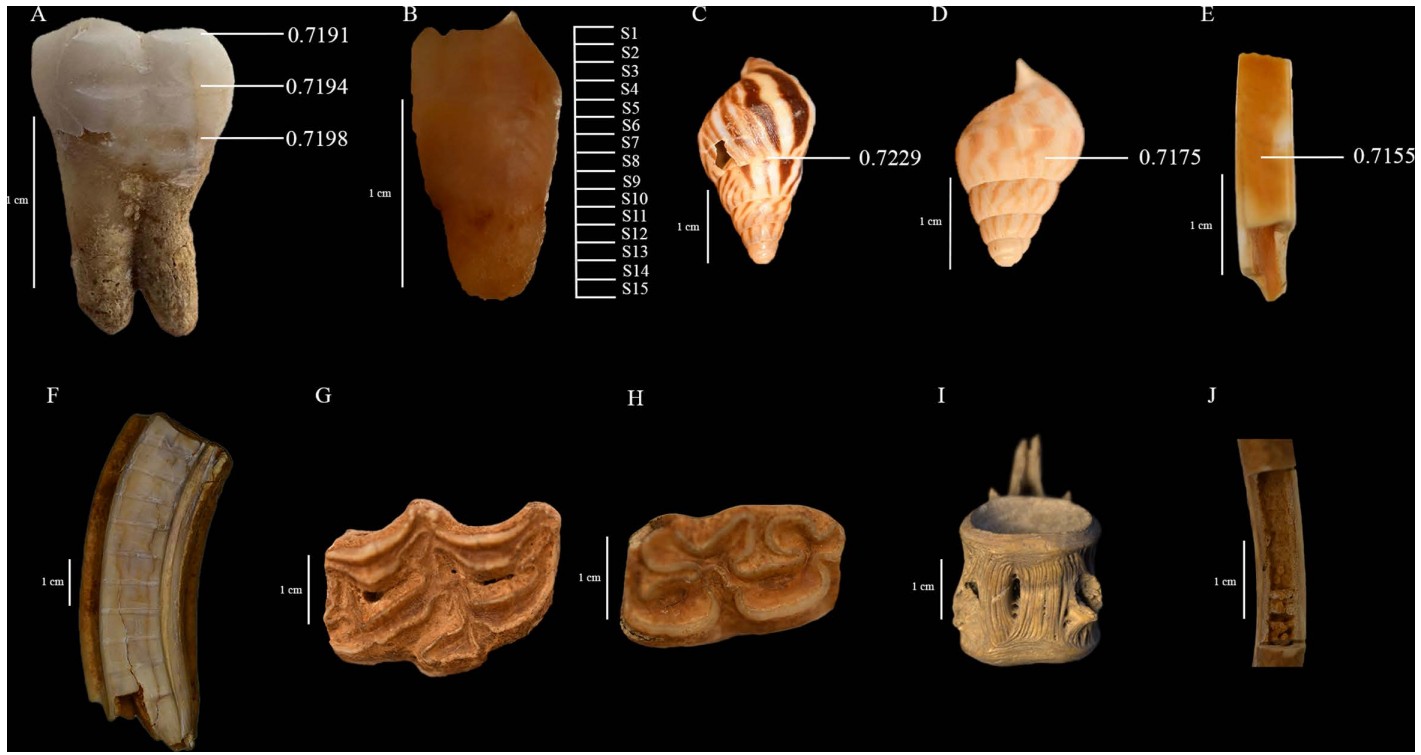

**Fig 6.** A – ⁸⁷Sr/⁸⁶Sr ratios from second upper molar of Individual 1 (*Homo sapiens*), B – dentinal collagen from Individual 1 showing the loci of sequential samples, C – ⁸⁷Sr/⁸⁶Sr ratios of snail shell (*Limicolaria flammea*) from Oyo-Ile (MJCV), D – ⁸⁷Sr/⁸⁶Sr ratios of snail shell (*Limicolaria flammea*) from Old Bara, E – ⁸⁷Sr/⁸⁶Sr ratios of cane rat (*Thryonomys swinderianus*) from Old Bara, F – horse (*Equus caballus*) tooth showing the multiple sampling loci, G – an upper dP3 from a foal (*Equus caballus*), H – lower molar of a donkey (*Equus asinus*), I – vertebrae of Nile perch (*Lates niloticus*), and J – femur of a Chicken (*Gallus gallus domesticus*) with dense medullary bone.

(Fig 9). Horse 1 would have spent at least a year in Old Bara (~20% of the tooth at the apical end was broken and was not sampled), while Horse 2 lived for about 34 months in Old Bara.

Horse 5 appears to be the only highly mobile individual among all horses analyzed and is originally not local to Old Bara or the capital (Figs 10 and 11). This individual would have been brought into Old Bara or through Oyo-Ile, but it spent the rest of its life in Old Bara. The alveolar bone of Horse 5 also has a $\delta^{34}$S value (Sample 27233) consistent with Old Bara. Horse 6 is also non-local and might have been brought into Old Bara (Fig 9). We also included Horse 7, the only horse tooth from the capital available during this analysis. Our probability map suggests that it is likely non-local or had spent most of its early life outside the capital and Old Bara (Fig 9). It is possible that Horses 5, 6, and 7 all came from the same location (Figs 9–11). The probability map also shows that the two donkey samples are local (Fig 12). We cannot ascertain whether these represent one individual, but they were residents in Old Bara for a substantial time of their early life (Fig 6H).

**$\delta^{13}$C, $\delta^{15}$N, $\delta^{34}$S and ⁸⁷Sr/⁸⁶Sr ratios of human remains.** The $\delta^{13}$C values from the sequential samples taken from the human molar (Individual 1) vary throughout the life of the individual (Fig 13). The standard deviation on $\delta^{13}$C from the molar is 2.22 ‰, with the most dramatic shift occurring between age two and a half ($\delta^{13}$C = −17.60 ‰) and about six and a half (−11.53 ‰). These correlate with a dietary switch from a nutritional regimen with high C$_3$ inputs toward a more C$_4$-dominant diet. This switch is likely associated with age-related dietary change between early and late childhood. There is no significant variation in the $\delta^{15}$N ($\sigma$ = 0.32 ‰) that would indicate a shift from the consumption of plant-based protein to

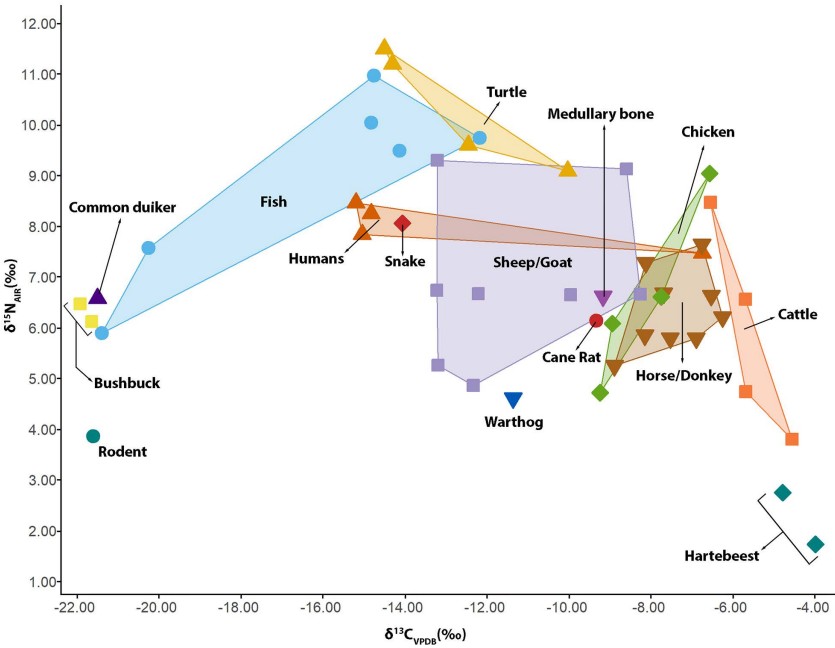

**Fig 7. A convex hull showing the carbon (δ¹³C) and nitrogen (δ¹⁵N) isotope composition of faunal and human remains analyzed.**

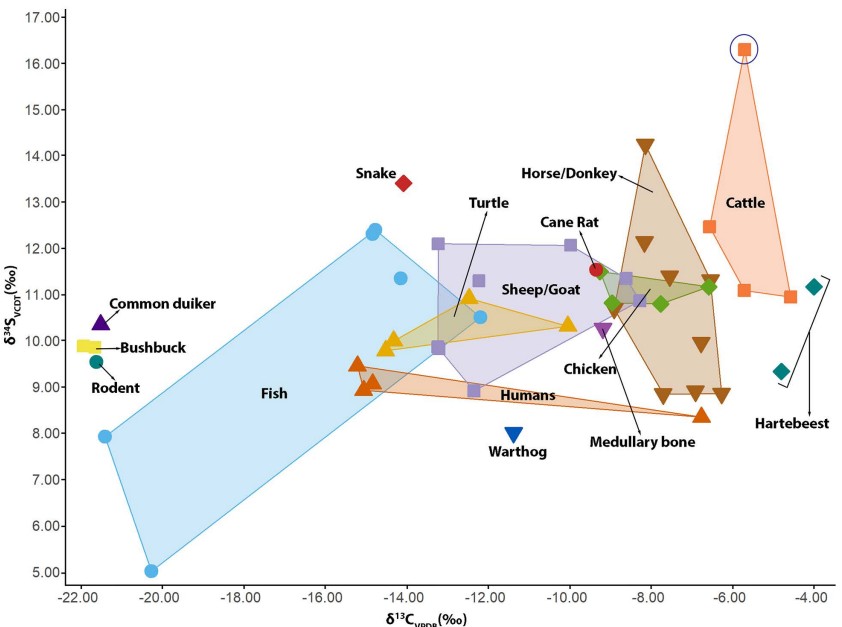

**Fig 8. A convex hull showing carbon (δ¹³C) and sulfur (δ³⁴S) isotope composition of faunal and human remains analyzed. Note: The cow/bull from a coastal environment is highlighted in blue circle.**

**Table 2. Mean and standard deviation for carbon (δ¹³C), nitrogen (δ¹⁵N), and sulfur (δ³⁴S) isotope values for faunal remains analyzed in the study (S3 Data).**

| Species | Common Name | MNI | $\delta^{13}C_{VPDB}$ (Mean±SD) | $\delta^{15}N_{AIR}$ (Mean±SD) | $\delta^{34}S_{VCDT}$ (Mean±SD) |
|---|---|---|---|---|---|
| *Equus* sp. | horse/donkey | 9 | −7.38±0.88 ‰ | +6.35±0.78 ‰ | +10.7±1.8 ‰ |
| *Bos taurus* | cattle | 4 | −5.58±0.82 ‰ | +5.89±2.07 ‰ | +12.7±2.5 ‰ |
| Caprinae | sheep/goat | 8 | −11.32±2.11 ‰ | +6.91±1.59 ‰ | +10.8±1.1 ‰ |
| *Phacochoerus africanus* | warthog | 1 | −11.31±0.34 ‰ | +4.61±0.27 ‰ | +8.0±0.2 ‰ |
| *Alcelaphus buselaphus* | hartebeest | 2 | −4.34±0.57 ‰ | +2.24±0.72 ‰ | +10.2±1.3 ‰ |
| *Tragelaphus scriptus* | harnessed bushbuck | 2 | −21.72±0.20 ‰ | +6.30±0.25 ‰ | +9.9±0.0 ‰ |
| *Sylvicapra grimmia* | common duiker | 1 | −21.4 ‰ | +6.6 ‰ | +10.3 ‰ |
| *Gallus domesticus* | chicken | 4 | −8.08±1.22 ‰ | +6.61±1.80 ‰ | +11.1±0.3 ‰ |
| Serpentes | snake | 1 | −14.0 ‰ | +8.1 ‰ | +13.4 ‰ |
| Actinopterygii | freshwater fish | 6 | −16.20±3.68 ‰ | +8.95±1.87 ‰ | +9.9±2.9 ‰ |
| Pelomedusidae | Side-necked turtle | 4 | −12.77±2.08 ‰ | +10.35±1.18 ‰ | +10.3±0.5 ‰ |
| *Thryonomys swinderianus* | cane rat | 1 | −9.3 ‰ | +6.1 ‰ | +11.5 ‰ |
| Rodentia | rodent indet. | 1 | −21.5 ‰ | +3.9 ‰ | +9.5 ‰ |

animal protein (Fig 13). There is also no significant variation in δ³⁴S values (σ=0.45 ‰), indicating that this individual may have had little to no residential mobility (Fig 13). In addition, the δ³⁴S values from this tooth fall within the range of the two rodents and the medullary bone used as a sulfur baseline for the site, further supporting the idea that this individual may have mainly lived within Old Bara. The mean values from the molar are: δ¹³C=−14.40 ‰, δ¹⁵N=8.65 ‰, and δ³⁴S = 10.16 ‰. These values are comparable to those bones from Individual 1 (talus, metatarsal, and calcaneus) recovered from the stratigraphic units below and above this tooth. Isotope values from a bone should provide a time-averaged representation of an individual's diet, and the consistency between bone and averaged tooth values suggests that the remains, with the exception of the cranial bone, belongs to Individual 1. The cranial bone has a much more positive carbon isotope composition (δ¹³C=−6.70 ‰) and lower δ¹⁵N and δ³⁴S values than other human samples (Fig 7 and 8). Although the cranial bone tends to have a very slow turnover rate, the high variation indicates that this bone is from a unique individual with a different dietary history [106].

Three enamel samples for strontium analysis were taken from the same human tooth at three locations (close to the occlusal surface, at the middle of the crown, and above the cementoenamel junction) with an interval of ~1.5 mm (Fig 6A). These samples should represent ages 2.5±0.5, 5.5±0.5, and 8.5±0.5, respectively [69]. The ⁸⁷Sr/⁸⁶Sr ratios from the molar ranges from 0.7191 to 0.7198, and the slight variation within the three values indicates that this individual had little residential mobility between the ages of 2 and 9. The ⁸⁷Sr/⁸⁶Sr ratios are consistent with δ³⁴S values obtained from the tooth. The ⁸⁷Sr/⁸⁶Sr ratios are similar to those of the local horses that suggests that this individual is local to Old Bara.

## Discussion

The presence of locally raised equines, caprines, cattle, and chickens in Old Bara suggests that the town was not solely dependent on supplies from the capital but was a thriving town with local production of resources and was part of the empire's distribution network of resources, especially in the metropolis. However, Old Bara also received some of its horses and cattle from outside the town. This includes the cow/bull (Sample 27223), that originated from a coastal environment, and the two horses (Horse 5 and 6), whose ⁸⁷Sr/⁸⁶Sr ratios indicate that they were previously non-local to Old Bara. These two horses (Horse 5 and 6) and the horse from Oyo-Ile (Horse 7) have ⁸⁷Sr/⁸⁶Sr ratios that are close to those from Nupe, northeastern Nigeria, and the Niger Delta. These horses might have come from Nupe (Figs 9 and 10). The

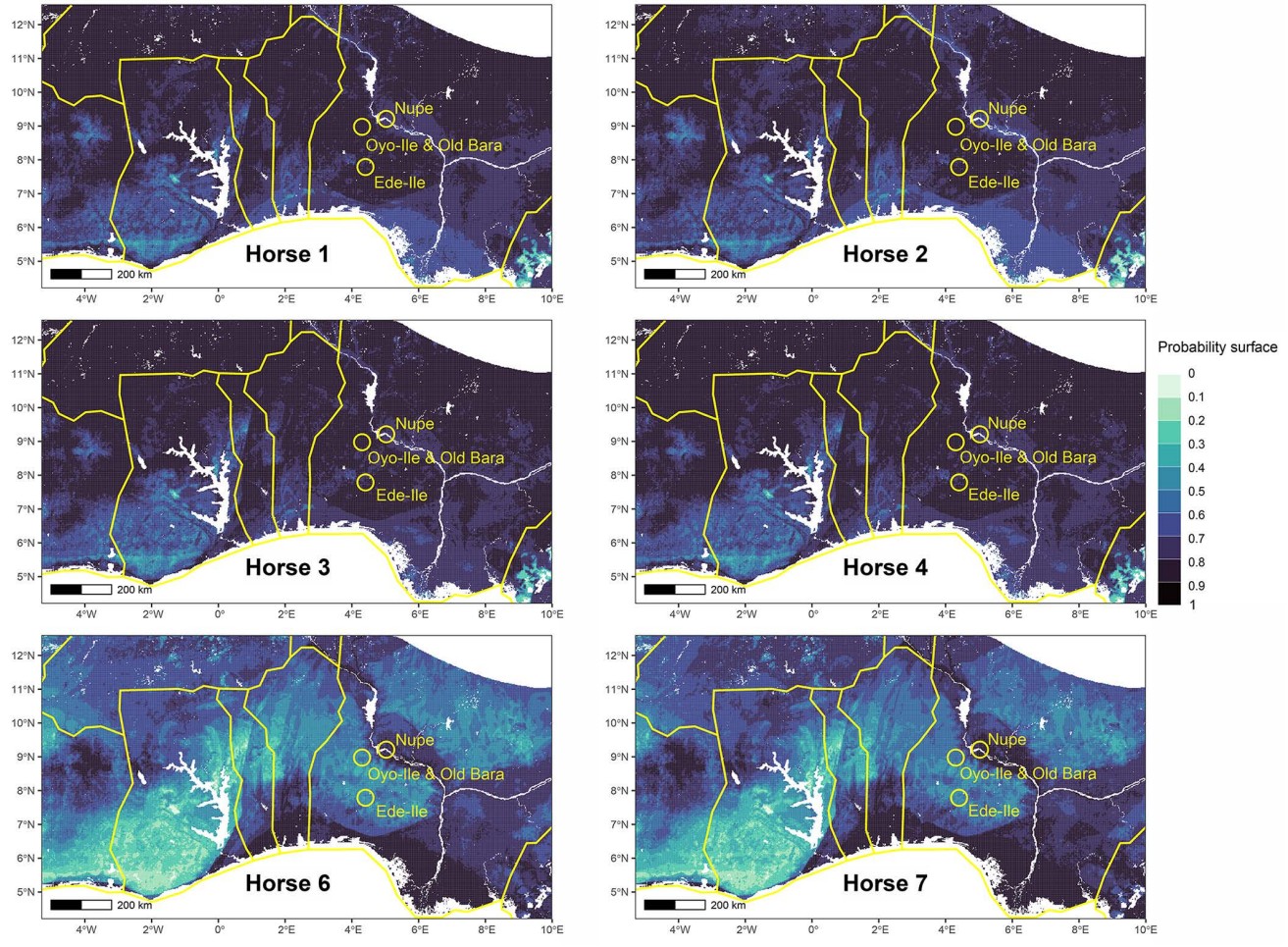

**Fig 9. Probability map of the places of origin for Horses 1, 2, 3, 4, 6, and 7 based on the range of West African Dwarf horses, highlighting the study area (Old Bara and Oyo-Ile) and two other cavalry centres, Ede-Ile and Nupe (Raba).**

Oyo Empire not only traded heavily with this northern neighbor, but until ca. 1780, Nupe was a vassal province of the empire.

Horses, cattle, and rams are important species sacrificed in Old Bara during the coronation of a new king and as part of their burial rituals [19]. Archaeological evidence suggests that the residents and special guests from the capital feasted on the sacrificed animals. Individuals participating in such feasts were prohibited from taking shares out of the town [19]. The people of Old Bara also had access to freshwater or wetland resources, such as fish (Fig 6I) and turtles. Some of these resources would have originated from outside the settlement or marshland south and southeast of the town. Wild species, such as hartebeest, harnessed bushbuck, common duiker, and warthog, would have been ultimately supplied to Old Bara via hunting beyond the Old Bara settlement landscape and their $\delta^{13}C$ values also supports that they came from different foraging environments.

The $\delta^{34}S$ and $^{87}Sr/^{86}Sr$ ratios from Individual 1 in this study indicate that the individual is probably local. The $\delta^{15}N$ and $\delta^{34}S$ values from Individuals 1 and 2 also do not show a substantial contribution of freshwater resources. The context of

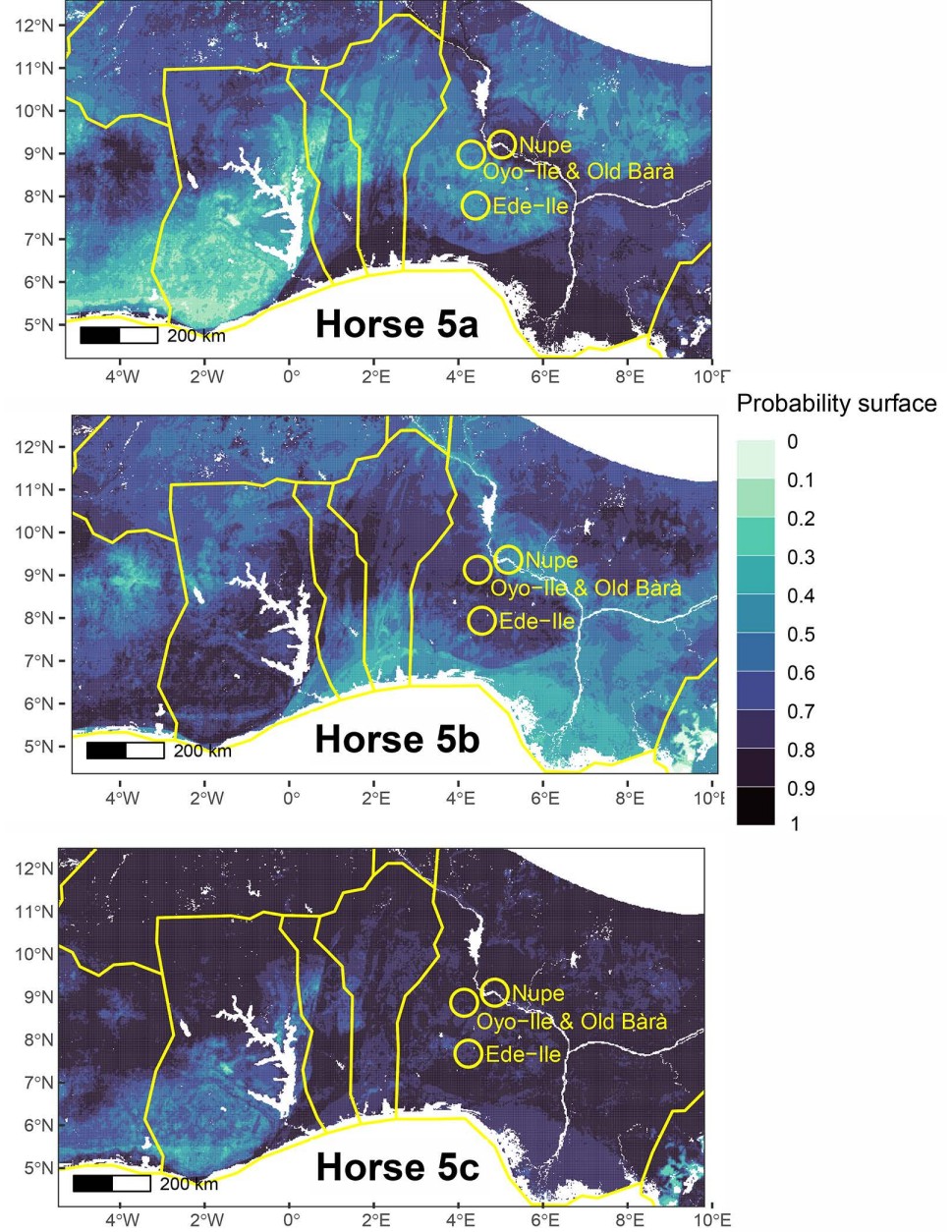

**Fig 10. Probability map of the places of origin for Horse 5 based on the range of West African Dwarf horses, highlighting the study area (Old Bara and Oyo-Ile) and two other cavalry centres, Ede-Ile and Nupe (Raba).**

their remains (samples likely swept into a refuse deposit) is indicative of their low status, as the burial of elites or kings within the royal site would have been sacred and protected from daily human activities.

In general, the $\delta^{13}C$ and $\delta^{15}N$ values from this study underscore the importance of cereals, which could be any or a combination of sorghum, millet, and maize, to the diet of humans, horses, cattle, and domestic chickens. Individual 2 with a more positive $\delta^{13}C$ value must have subsisted heavily on a $C_4$ diet. The importance of cereal in the diet of people in Old

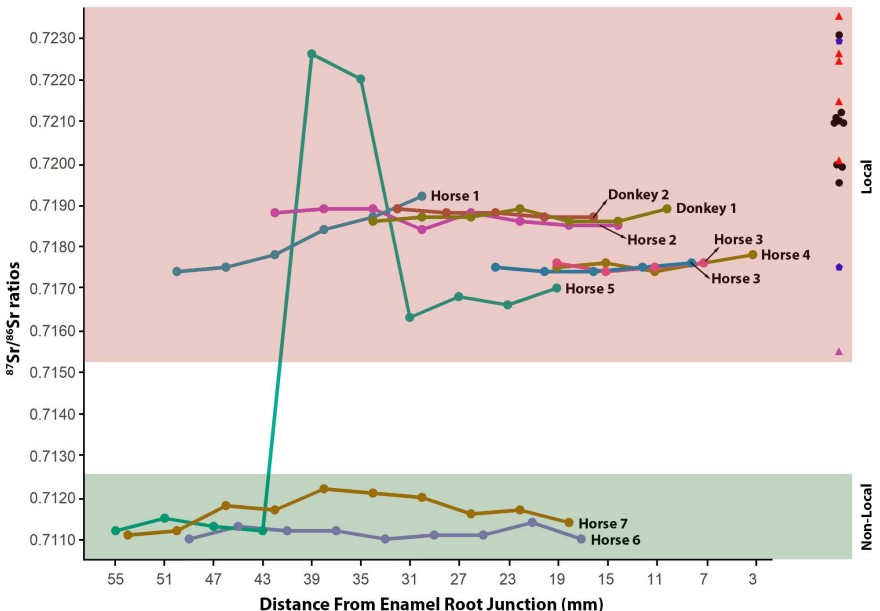

**Fig 11. Summary of ⁸⁷Sr/⁸⁶Sr ratios for horses and donkeys. The X-axis represents sequential samples from occlusal to enamel-root junction (black circles=cementum leachates, red triangles=soil leachates, magenta triangle=cane rat, and blue pentagon=snail shell).**

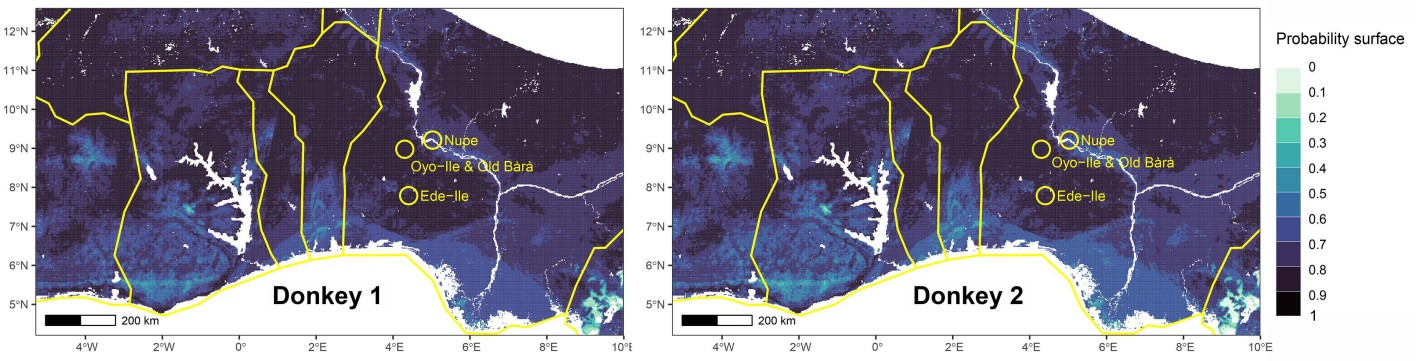

**Fig 12. Probability map of the places of origin for Donkey 1 and 2 based on the range of West African Dwarf horses, highlighting the study area (Old Bara and Oyo-Ile) and two other cavalry centres, Ede-Ile and Nupe (Raba).**

Bara also aligns with isotope data from Elmina, coastal Ghana, in the early 17th century [107]. Caprines could have been provided with stovers and straws from cereals, but culturally, they are generally allowed to graze around compounds.

Given that horses were being bred in Old Bara, this site may have been one of the breeding grounds for the Oyo Empire light cavalry, as horse breeding was reported to have taken place in the countryside [23]. Old Bara, located in the suburb of Oyo-Ile fits this model, and would have supported the capital with some of its WAD horses. However, horses were not exclusively for cavalry; they served multiple functions, including ritual feasting, transportation, and ceremonial purposes, and may be slaughtered for food when they become feeble or sick. The latter is well supported by the cut marks on the postcranial bones of horses from the site. Donkeys, as the beast of burden, would have also contributed to the

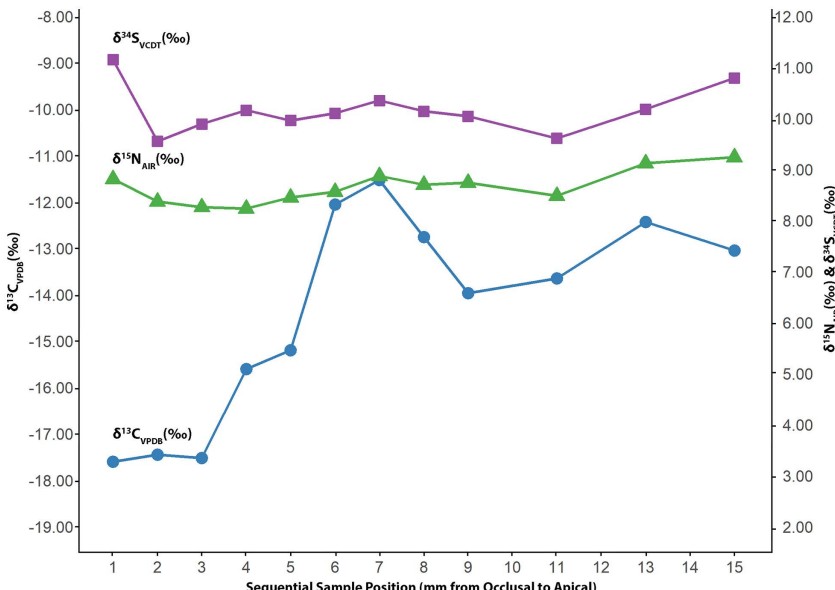

**Fig 13. Carbon ($\delta^{13}$C), nitrogen ($\delta^{15}$N), and sulfur ($\delta^{34}$S) isotope values of the tooth from Individual 1 with significant variation in $\delta^{13}$C and little variation in $\delta^{15}$N and $\delta^{34}$S.**

movement of resources within and outside the town, and the $^{87}$Sr/$^{86}$Sr ratios from this species suggest that donkeys were likely being raised and used locally in Old Bara.

The more positive $\delta^{13}$C values of horses in Old Bara fall within the range of sorghums (*Sorghum arundinaceum* and *Sorghum bicolor*) and millets (*Pennisetum glaucum* and *Eleusine coracana*) from West Africa and the expected values for maize. The isotope data presented in this study support historical sources indicating that grasses and cereals, particularly great millet (*Sorghum bicolor*) and Burgu millet (*Echinochloa stagnina*), were the primary diet of horses in West Africa. Although Robin Law noted that cereals were often complemented with legumes such as beans and groundnuts [23], the more positive $\delta^{13}$C values from these horses suggest that it is unlikely equines were being fed $C_3$ plants like beans and groundnuts. Foddering with groundnuts could have been a common practice further north in the open savanna, where they are widely grown. Quantifying or discriminating the contribution of sorghum, millet, and maize to animal diets in this environment is almost impossible because $C_4$ plants exhibit much less variation than $C_3$ plants due to their carbon-concentrating mechanisms and limited opportunities for fractionation.

During dry seasons, access to stovers and straws from cereals and grasses becomes problematic for horse keepers. Officers in charge of stables depended on enslaved people to acquire grasses, particularly Burgu millet (*Echinochloa stagnina*), for feeding the horses. These grasses were also collected as tribute from vassal states [23]. Burgu millet is a $C_4$ plant that can be easily accessed from the marshes of the River Niger, 40–50 kilometers from Oyo-Ile and Old Bara. This supply of fodder would have been sourced directly from the wetlands of the River Niger or as tributes from vassal towns and cities outside the metropolis, especially in the northern extremes of the empire. The fodder could also have been purchased from the drier northern region. This non-local dry season fodder likely accounts for the wide range seen in $\delta^{34}$S values of horses and the grouping or variations observed in the $^{87}$Sr/$^{86}$Sr ratios of local horses. These factors should be considered when interpreting causality and variability in isotope systems, such as strontium, from extractive hegemonies and trading empires with histories of cavalry or horsemanship, as was the case in the Oyo Empire and other West African polities, such as Mali, Songhai, and Kanem-Bornu.

## Critical considerations for provenance and mobility studies within the Oyo Empire

Several factors may limit the use of strontium and sulfur isotopes in tracing geographic origins and mobility patterns within the Oyo Empire and similar polities in West Africa. The most important consideration when conducting provenance studies in this area is understanding the structural complexities of the Oyo Empire. The empire was a cosmopolitan state that mobilized people and resources from different parts of West Africa [12,108]. As also argued above, tributes were an important component of the capital's resources. The centers of power, such as Oyo-Ile and Old Bara, subsisted through local production, tributes, and gifts from colonies and vassal states. The consumption of non-local resources is often a limiting factor when using strontium and sulfur for mobility studies. However, once identified, these practices can provide valuable information on resource redistribution and exchange [64]. This problem is not limited to human dietary reconstruction but also includes other non-human animals. In West Africa, for example, fodders were often traded and imposed as tributes to colonies, particularly during the dry season when grasses were scarce and unavailable [23]. This is particularly important for the Oyo Empire's metropolis, where horses were central to the empire's political structure and military might.

The trade and consumption of salt during this period present yet another concern when using strontium for mobility studies. The Oyo Empire obtained sea salt and rock salt through trade and tributes from the coastal areas and through trade from their northern neighbors as far as Tripoli [16,25]. This trade implies that the local $^{87}Sr/^{86}Sr$ ratios in humans and animals consuming imported sea or rock salt could be masked when consumed in significant quantities [109], particularly when the imported salts have a different $^{87}Sr/^{86}Sr$ ratios to those of the local environment [110,111]. Similarly, the Yoruba add potash to their food as a preservative, tenderizer, or to increase the viscosity of their soups. In addition, cattle are often provided with salt to enhance water conservation and disease resistance, and people often add potash as an important supplement to horse fodder [23]. These practices could alter the $^{87}Sr/^{86}Sr$ ratios of animals and humans. Based on this, local species and individuals could be misidentified as non-locals, and the overall $^{87}Sr/^{86}Sr$ ratios in species could be altered in a way that makes mobility studies difficult. Lastly, the Oyo Empire was highly stratified, and class differentiation would have affected the diets of both elites and commoners. For example, sumptuary laws restricted the consumption of horse meat to the elites and their dependents. This could make the isotopic composition of those individuals distinct from the non-elite or those with limited access to elite privileges. In this study, we do not see evidence of significant marine salt consumption by either humans or equines, as their $^{87}Sr/^{86}Sr$ ratios deviate from that of ocean water.

The least concerning factor is the lack of a fine-grained isoscape map for important centers and colonies of the Oyo Empire. This challenge has been ameliorated by a recent Sr isoscape publication for sub-Saharan Africa [77], which is adapted for this study. The Sr isoscape has significantly improved our understanding of variations in $^{87}Sr/^{86}Sr$ ratios on a regional scale. However, the degree of variation on a more local scale across geospatial areas in southwestern Nigeria, Benin, and Togo that the Oyo Empire once controlled needs refinement.

A possible solution to some of the above challenges would be to apply multiproxy stable isotope analysis, as done in this study. This approach provides multiple lines of evidence that may help untangle some of these complexities. For example, while salt can average out and alter the $^{87}Sr/^{86}Sr$ ratios in humans and animals, sulfate from imported or locally sourced salt and potash would have a minimal effect on the $\delta^{34}S$ values of herbivores and humans, since herbivores cannot metabolize inorganic forms of sulfur. In addition, we can quickly use $\delta^{15}N$ values to identify individuals whose $^{87}Sr/^{86}Sr$ ratios have been altered by sea salt, provided they do not consume marine or freshwater resources. Sulfur is also an alternative isotope system for cattle and horses whose $^{87}Sr/^{86}Sr$ ratios might have been altered through imported supplementary salt or potash. The $\delta^{34}S$ values of these species should only reflect those of marine environments if they grazed in an area that receives considerable sea spray or have been fed with plants from environments with marine algae.

Informed by these complexities, this paper has offered the first stable isotope and ZooMS analyses via collagen in Nigeria to begin to answer questions about human and animal mobility, diet, residential attachment to place, and animal/plant resource distribution in the Oyo Empire by focusing on Old Bara, an important place in the settlement geography of

the empire's metropolis. This paper sets the stage for further study by providing frameworks for exploring questions about human and animal diets, origins, and the management of animals, especially equines, in West Africa.

## Conclusion

Old Bara was not a passive settlement of palace retirees, celibate priests and priestesses attending to the temple of the Oyo kings, that only depended on provisions from the capital. The biogeochemistry of the fauna assemblage suggests that it was an integral part of the metropolitan political economy, engaging in local production, serving as a source of supply, and participating in the redistribution network of tributes and commercial goods originating from far-flung areas across the empire. Contrary to the impression in the historical records that only adults and celibates were residents of Old Bara, isotope data from the human remains suggests that children were also raised and might even have been born in this suburban royal town where rituals of coronation and burial supposedly took place, and where several palace officials were relocated to serve as priests and priestesses of the deceased and deified kings.

## Supporting information

**S1 Data. Supporting figures and ZooMS report.**
(DOCX)

**S2 Data. Inclusivity in global research questionnaire.**
(DOCX)

**S3 Data. Carbon ($\delta^{13}$C), nitrogen ($\delta^{15}$N), and sulfur ($\delta^{34}$S) isotope data for all analyzed samples.**
(XLSX)

**S4 Data. Strontium isotope ratios ($^{87}$Sr/$^{86}$Sr) for all analyzed samples.**
(XLSX)

## Acknowledgments

We extend our appreciation to all members of the Oyo Empire Archaeological Research Project from 2017 to 2022, as well as to the Nigerian National Park Service officers and staff, for their logistical support. We gratefully acknowledge that Nigeria's National Commission for Museums and Monuments enthusiastically granted the permit for the fieldwork and export of the samples. We thank Matt Teeter and Karla Newman for their technical assistance during the isotope analysis, Samantha Greeves at the University of York, UK, for assistance with MALDI-TOF MS, the York Centre of Excellence in Mass Spectrometry for MALDI-TOF access, the Royal Ontario Museum (ROM), Jacqueline Miller, and Wim Van Neer for their assistance during the faunal analysis presented in this study, and Mech E. Frazier for preparing the Oyo Empire and Nigerian geology maps. Finally, we thank all the anonymous reviewers whose suggestions helped improve this article.

## Author contributions

**Conceptualization:** Moses Oluwaseyi Akogun, Akinwumi Ogundiran.

**Data curation:** Moses Oluwaseyi Akogun, Paul Szpak.

**Formal analysis:** Moses Oluwaseyi Akogun, Paul Szpak, Vicky M. Oelze, Kendra Leishman, Jay Hilsden, Camilla Speller.

**Funding acquisition:** Moses Oluwaseyi Akogun, Paul Szpak, Camilla Speller, Akinwumi Ogundiran.

**Investigation:** Moses Oluwaseyi Akogun, Jonathan O. Aleru, Akinwumi Ogundiran.

**Methodology:** Moses Oluwaseyi Akogun.

**Supervision:** Paul Szpak.

**Visualization:** Moses Oluwaseyi Akogun, Vicky M. Oelze.

**Writing – original draft:** Moses Oluwaseyi Akogun, Akinwumi Ogundiran, Camilla Speller.

**Writing – review & editing:** Paul Szpak, Lisa Janz, Akinwumi Ogundiran.

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
