## [Decision Letter · Decision Letter 0]

9 Feb 2026

Dear Dr. Akogun,

Thank you for submitting your manuscript to PLOS ONE. After careful consideration, we feel that it has merit but does not fully meet PLOS ONE’s publication criteria as it currently stands. Therefore, we invite you to submit a revised version of the manuscript that addresses the points raised during the review process.

We look forward to receiving your revised manuscript.

Kind regards,

Enrico Greco

Academic Editor

PLOS One

Journal Requirements:

https://journals.plos.org/plosone/s/file?id=ba62/PLOSOne_formatting_sample_title_authors_affiliations.pdf....

3. In your manuscript, please provide additional information regarding the specimens used in your study. Ensure that you have reported human remain specimen numbers and complete repository information, including museum name and geographic location.

For more information on PLOS One's requirements for paleontology and archeology research, see https://journals.plos.org/plosone/s/submission-guidelines#loc-paleontology-and-archaeology-research....

“The fieldwork was funded by a National Geographic Explorers Grant NGS-63339R-19 (A.O), an AIA-NEH Grant for Archaeological Research (A.O), and a University of North Carolina at Charlotte Faculty Research Grant (A.O). The analysis was supported via funding from the Canada Research Chairs program, NSERC Discovery RTI (2023-00124) (P.S), NSERC Discovery Grant (RGPIN-2019-04145) (C.S), and Northwestern University’s Cardiss Collins Professorship (A.O).”

“The fieldwork was funded by a National Geographic Explorers Grant NGS-63339R-19 (A.O), an AIA-NEH Grant for Archaeological Research (A.O), and a University of North Carolina at Charlotte Faculty Research Grant (A.O). The analysis was supported via funding from the Canada Research Chairs program, NSERC Discovery RTI (2023-00124) (P.S), NSERC Discovery Grant (RGPIN-2019-04145) (C.S), and Northwestern University’s Cardiss Collins Professorship (A.O).”

6. Please be informed that funding information should not appear in the Acknowledgments section or other areas of your manuscript. We will only publish funding information present in the Funding Statement section of the online submission form. Please remove any funding-related text from the manuscript.

7. We note that Figures 1,2,3 & 5 in your submission contain [map/satellite] images which may be copyrighted. All PLOS content is published under the Creative Commons Attribution License (CC BY 4.0), which means that the manuscript, images, and Supporting Information files will be freely available online, and any third party is permitted to access, download, copy, distribute, and use these materials in any way, even commercially, with proper attribution. For these reasons, we cannot publish previously copyrighted maps or satellite images created using proprietary data, such as Google software (Google Maps, Street View, and Earth). For more information, see our copyright guidelines: http://journals.plos.org/plosone/s/licenses-and-copyright.

a. You may seek permission from the original copyright holder of Figures 1,2,3 & 5 to publish the content specifically under the CC BY 4.0 license.

Reviewers' comments:

Reviewer's Responses to Questions

**Comments to the Author**

1. Is the manuscript technically sound, and do the data support the conclusions?

Reviewer #1: Yes

Reviewer #2: Yes

2. Has the statistical analysis been performed appropriately and rigorously?

Reviewer #1: Yes

Reviewer #2: Yes

3. Have the authors made all data underlying the findings in their manuscript fully available?

Reviewer #1: Yes

Reviewer #2: Yes

4. Is the manuscript presented in an intelligible fashion and written in standard English?

Reviewer #1: Yes

Reviewer #2: Yes

Reviewer #1: This interdisciplinary study presents some interesting results on faunal and human remains from Nigeria. The text is well written and well structured. The number of human individuals included in this study is low, and more precaution should be taken when the authors generalize the results to produce a population-level interpretation. I believe the nature of the samples chosen to create the strontium baseline is not unanimously approved, and could benefit from being further discussed and argumented by the authors. I agree with the conclusions of the authors regarding the general socio-economic interpretation of the data.

The numbers in front of the comments refer to the lines of the manuscript.

177-179. Is there a reason why this multi-isotope study does not include oxygen isotope analysis, which are very commonly performed alongside strontium isotope analysis in mobility studies? If this is a choice that derives from the initial design of this study, it should be mentioned and explained here.

245-246 Citation needed.

247-249 You should mention why these values are of interest for the reader.

330-333 I assume you analyzed horse cementum either because it was formed while the animals were living at the site, or because it is thought to be more susceptible to diagenesis and would thus represent the local Sr isotopic signature of the soil. Either way, you have to explain why you analyzed this type of tissue to look at variability, as I do not think you mention cementum in the general paragraph about isotopic analyses.

Furthermore, I do not know of any study performing strontium isotope analysis on cementum of archaeological material. If you know of any, it should be cited here. The general consensus is that tooth enamel should be prefered over bones when it comes to strontium isotope analysis of archaeological material, as the latter are more susceptible to suffer from diagenesis. Cementum contains more organic material than enamel, and, therefore, could also be degraded by diagenetic processes. I think you should address these issues here.

401-404 I assume some of these samples are modern and other are from archaeological material; it should be made explicit. The use of snail shells for establishing a local baseline has been criticized recently, it may be good to acknowledge it.

404-408 You should indicate here why you are using soil samples for your baseline, while you also use a probability map that is explicitly based on organic material in order to represent bioavailable strontium signatures over this area. Many authors claim that the relationship between the strontium ratio found in the soils and the bioavailable strontium that characterizes a local system is not strictly linear and can be influenced by a certain number of factors, so it would be useful to acknowledge this fact here.

477 The values are compatible with a local origin, but I do not think they constitute a clear evidence- they could also be consistent with an origin from another place with a similar strontium ratio. It would be better to rephrase.

563-565 Their remains may also have been moved after the site had been abandonned.

629-630 I think this sentence is not very clear- would the local ratio be altered by small quantities of imported salt, or by significant quantities of it? Or unaltered even by significant consumption of salt? That would depend on whether the strontium ratio of the imported foodstuff is different from those locally available, in fine.

626-642 A few studies have tried to provide a quantitative assessment of the amount of salt needed to alter the local strontium ratio of humans, see Fenner 2014 for instance.

658-660 Recent studies also suggest local variations exist within terrestrial landscapes, for instance see Tarrant and Richards 2024.

Fig. 5 You should also include the location of the site and the soil samples on the geological map.

Reviewer #2: The article is an important addition to the literature where there are few written and oral records about the Old Bara site. The authors capture how Archaeology, Zooarchaeology and Bioarchaeology studies have afforded researchers an opportunity to explore past life in the Oyo Empire and its entirety. The abstract and the introduction coherently summarises the article’s research question, key findings, and how the work fits in the existing West African research. The contextualisation of the Oyo Empire and the Old Bara create a baseline to understand/appreciate the authors research investigating the diet of animals and humans at Old Bara and the source of the resources & production centres. To gain these insights into residential mobility, diet and animal management, the authors employ a multiproxy stable isotope ( 13C, 15N, 34S, 87Sr/86Sr), zooarchaeology and Zooarchaeology by Mass Spectrometry (ZooMS) analyses to human and faunal remains excavated and collected from Old Bara between 2018 and 2022. The robust approach of assessing multiple proxies fits the research question and capturing as much data as possible from the limited study material they had available. The authors included methodology backgrounds which shows how they each fit their overall research and the methods are well explained. The tables and the figures align well with the text, the results are well discussed together with the possible limitations and how those shape up the interpretation/manipulation of the data generated. Given the data generated the statistics included in the article is fairly sufficient. A few things the authors could consider are:

- Adding a table that summarises the materials, number and which analysis they were used for (a summary of Pg 13, line 267-271).

- Check “Mores” from Pg 14, line 294, it might be Moorrees.

- Adding a few sub-headings in the methods sections to make it more legible and easier to read.

- Adding points on the Nigerian geological map (figure 5) to clearly show the location of the mentioned 87Sr/86Sr rations.

- The statement on Pg 25, line 522-524 is a bit unclear or incomplete, it might need rephrasing if it is to conclude that “the consistency between bone and tooth values suggests that the remains, with exception of the cranial bone belong to one individual” being individual 1 not 2.

In conclusion the article is well written and shows the extensive research that has been put through to test the historical and oral records with evidence-based results. I believe the article does advance the zooarchaeological, bioarchaeological and biogeochemical fields in understanding past communities in West Africa.

.

Reviewer #1: No

Reviewer #2: No

---

## [Author Response · Author response to Decision Letter 1]

6 Mar 2026

On behalf of my coauthors, I would like to thank you and the anonymous reviewers for finding merit in our manuscript entitled “Multiproxy stable isotope analysis provides insights into diet, animal management, and residential mobility in Old Bara, a metropolitan suburb of the Old Oyo Empire”. Please find below responses to the issues raised by the academic editor and reviewers.

In addition to the request made below based on the comment from the reviewers and the editorial team, we also request that:

1. The editorial team change the title of the manuscript to: “Multiproxy stable isotope analysis provides insights into diet, animal management, and residential mobility in Old Bara, a metropolitan suburb of the Oyo Empire, West Africa.”

This helps to highlight the region where the empire was and also by removing “Old” from “Old Oyo”, the title remains consistent with the nomenclature used across the manuscript.

2. The editorial team add “Archaeology”, “Zooarchaeology” and “West Africa” to the keywords so that it reads: Keywords: Archaeology, Old Bara, animal management, residential mobility, Zooarchaeology, ZooMS, stable isotope analysis, West Africa.

These changes have been added as Track Changes to the revised manuscript along with a few edits to improve clarity, consistency and flow of the writing.

Journal Requirements:

1. Please ensure that your manuscript meets PLOS ONE's style requirements, including those for file naming. The PLOS ONE style templates can be found at https://journals.plos.org/plosone/s/file?id=wjVg/PLOSOne_formatting_sample_main_body.pdf and https://journals.plos.org/plosone/s/file?id=ba62/PLOSOne_formatting_sample_title_authors_affiliations.pdf.

Response: The manuscript has been formatted in accordance with PLOS One’s guidelines for the main body of the manuscript, including those related to the title, authors, and affiliations. Thank you.

Response: Thank you. We have completed the PLOS questionnaire on inclusivity in global research and have cited it in the manuscript. The completed copy has been attached as Supplementary 2 (S2).

3. In your manuscript, please provide additional information regarding the specimens used in your study. Ensure that you have reported human remain specimen numbers and complete repository information, including museum name and geographic location.

For more information on PLOS One's requirements for paleontology and archeology research, see https://journals.plos.org/plosone/s/submission-guidelines#loc-paleontology-and-archaeology-research.

Response: Thank you. Catalog and context IDs for all human and animal remains are provided in Supplementary 3 (S3). A statement confirming that all necessary permits were obtained has also been added to the manuscript.

“The fieldwork was funded by a National Geographic Explorers Grant NGS-63339R-19 (A.O), an AIA-NEH Grant for Archaeological Research (A.O), and a University of North Carolina at Charlotte Faculty Research Grant (A.O). The analysis was supported via funding from the Canada Research Chairs program, NSERC Discovery RTI (2023-00124) (P.S), NSERC Discovery Grant (RGPIN-2019-04145) (C.S), and Northwestern University’s Cardiss Collins Professorship (A.O).”

Response: Thank you. We request that the editorial team update the funding statement with the following:

“The fieldwork was funded by a National Geographic Explorers Grant NGS-63339R-19 (A.O), an AIA-NEH Grant for Archaeological Research (A.O), and a University of North Carolina at Charlotte Faculty Research Grant (A.O). The analysis was supported via funding from the Canada Research Chairs program (P.S), NSERC Discovery RTI (2023-00124) (P.S), NSERC Discovery Grant (RGPIN-2019-04145) (C.S), Northwestern University’s Cardiss Collins Professorship (A.O), Connaught International Scholarship for Doctoral Students (M.O.A), Ontario Graduate Scholarship (M.O.A), and Vanier Canada Graduate Scholarships (FRN: 198885) (M.O.A). There was no additional external funding received for this study.”

“The fieldwork was funded by a National Geographic Explorers Grant NGS-63339R-19 (A.O), an AIA-NEH Grant for Archaeological Research (A.O), and a University of North Carolina at Charlotte Faculty Research Grant (A.O). The analysis was supported via funding from the Canada Research Chairs program, NSERC Discovery RTI (2023-00124) (P.S), NSERC Discovery Grant (RGPIN-2019-04145) (C.S), and Northwestern University’s Cardiss Collins Professorship (A.O).”

Response: Thank you. We request editorial team update the financial disclosure with the following:

“The funders had no role in study design, data collection and analysis, decision to publish, or preparation of the manuscript”

6. Please be informed that funding information should not appear in the Acknowledgments section or other areas of your manuscript. We will only publish funding information present in the Funding Statement section of the online submission form. Please remove any funding-related text from the manuscript.

Response: Thank you. Information about funding statement has been removed from the manuscript.

7. We note that Figures 1,2,3 & 5 in your submission contain [map/satellite] images which may be copyrighted. All PLOS content is published under the Creative Commons Attribution License (CC BY 4.0), which means that the manuscript, images, and Supporting Information files will be freely available online, and any third party is permitted to access, download, copy, distribute, and use these materials in any way, even commercially, with proper attribution. For these reasons, we cannot publish previously copyrighted maps or satellite images created using proprietary data, such as Google software (Google Maps, Street View, and Earth). For more information, see our copyright guidelines: http://journals.plos.org/plosone/s/licenses-and-copyright.

a. You may seek permission from the original copyright holder of Figures 1,2,3 & 5 to publish the content specifically under the CC BY 4.0 license.

Response: Thank you. Images made with data from Google software have been removed and the sources of all other images were cited accordingly.

Response: Thank you. We have incorporated most of the references recommended by the reviewers.

Response: Thank you. The following ten references were added to the reference list, and the reference list has updated and formatted according to PLOS One requirements. The first two references were added to the reference list because we omitted them while reporting the C14 dates and the last eight were added based on the comments and recommendations of the reviewers.

Ramsey CB. OxCal v4. 4.4. 2021.

Reimer PJ, Austin WEN, Bard E, Bayliss A, Blackwell PG, Bronk Ramsey C, et al. The IntCal20 Northern Hemisphere Radiocarbon Age Calibration Curve (0–55 cal kBP). Radiocarbon. 2020;62(4):725-57. doi: 10.1017/RDC.2020.41.

James HF, Adams S, Willmes M, Mathison K, Ulrichsen A, Wood R, et al. A large-scale environmental strontium isotope baseline map of Portugal for archaeological and paleoecological provenance studies. Journal of Archaeological Science. 2022;142:105595. doi: https://doi.org/10.1016/j.jas.2022.105595.

Adams S, Grün R, McGahan D, Zhao J-X, Feng Y, Nguyen A, et al. A strontium isoscape of north-east Australia for human provenance and repatriation. Geoarchaeology. 2019;34(3):231-51. doi: https://doi.org/10.1002/gea.21728.

Warham JO. Mapping biosphere strontium isotope ratios across major lithological boundaries. A systematic investigation of the major influences on geographic variation in the 87Sr/86Sr composition of bioavailable strontium above the Cretaceous and Jurassic rocks of England: University of Bradford; 2013.

Hartman G, Richards M. Mapping and defining sources of variability in bioavailable strontium isotope ratios in the Eastern Mediterranean. Geochimica et Cosmochimica Acta. 2014;126:250-64. doi: 10.1016/j.gca.2013.11.015.

Evans JA, Montgomery J, Wildman G, Boulton N. Spatial variations in biosphere 87Sr/86Sr in Britain. Journal of the Geological Society. 2010;167(1):1-4.

Maurer A-F, Galer SJG, Knipper C, Beierlein L, Nunn EV, Peters D, et al. Bioavailable 87Sr/86Sr in different environmental samples — Effects of anthropogenic contamination and implications for isoscapes in past migration studies. Science of The Total Environment. 2012;433:216-29. doi: https://doi.org/10.1016/j.scitotenv.2012.06.046.

Pryor AJ, Ameen C, Liddiard R, Baker G, Kanne KS, Milton JA, et al. Isotopic biographies reveal horse rearing and trading networks in medieval London. Science Advances. 2024;10(12):eadj5782.

Fenner JN, Wright LE. Revisiting the strontium contribution of sea salt in the human diet. Journal of Archaeological Science. 2014;44:99-103.

Review Comments to the Author

Reviewer #1: This interdisciplinary study presents some interesting results on faunal and human remains from Nigeria. The text is well written and well structured. The number of human individuals included in this study is low, and more precaution should be taken when the authors generalize the results to produce a population-level interpretation. I believe the nature of the samples chosen to create the strontium baseline is not unanimously approved, and could benefit from being further discussed and argumented by the authors. I agree with the conclusions of the authors regarding the general socio-economic interpretation of the data.

177-179. Is there a reason why this multi-isotope study does not include oxygen isotope analysis, which are very commonly performed alongside strontium isotope analysis in mobility studies? If this is a choice that derives from the initial design of this study, it should be mentioned and explained here.

Response: Thank you for the comment. We did not include oxygen isotope analysis in the research design as our laboratory is not yet set up for oxygen isotope analysis.

245-246 Citation needed.

Response: Thank you, a citation has been provided for the statement.

247-249 You should mention why these values are of interest for the reader.

Response: Thank you. This has been explained in the revised manuscript.

330-333 I assume you analyzed horse cementum either because it was formed while the animals were living at the site, or because it is thought to be more susceptible to diagenesis and would thus represent the local Sr isotopic signature of the soil. Either way, you have to explain why you analyzed this type of tiss

---

## [Editor Report · Decision Letter 1]

12 Mar 2026

Multiproxy stable isotope analysis provides insights into diet, animal management, and residential mobility in Old Bara, a metropolitan suburb of the Oyo Empire, West Africa

PONE-D-25-64727R1

Dear Dr. Akogun,

We’re pleased to inform you that your manuscript has been judged scientifically suitable for publication and will be formally accepted for publication once it meets all outstanding technical requirements.

Kind regards,

Enrico Greco

Academic Editor

PLOS One
---

## [Editor Report · Acceptance letter]

PONE-D-25-64727R1

PLOS One

Dear Dr. Akogun,

I'm pleased to inform you that your manuscript has been deemed suitable for publication in PLOS One. Congratulations! Your manuscript is now being handed over to our production team.

Kind regards,

on behalf of

Dr. Enrico Greco

Academic Editor

PLOS One